# Distributed SGD in overparameterized Linear Regression

## Abstract

We consider distributed learning using constant stepsize SGD over several devices, each sending a final model update to a central server. In a final step, the local estimates are aggregated. We prove in the setting of overparameterized linear regression general upper bounds with matching lower bounds and derive learning rates for specific data generating distributions. We show that the excess risk is of order of the variance provided the number of local nodes grows not too large with the global sample size.

We further compare distributed SGD with distributed ridge regression and provide an upper bound of the excess SGD-risk in terms of the excess RR-risk for a certain range of the sample size.

## 1 INTRODUCTION

Deep neural networks possess powerful generalization properties in various machine learning applications, despite being overparameterized. It is generally believed that the optimization algorithm itself, e.g., stochastic gradient descent (SGD), implicitly regularizes such overparameterized models. This regularizing effect due to the choice of the optimization algorithm is often referred to as *implicit regularization*. A refined understanding of this phenomenon was recently gained in the setting of linear regression (to be considered as a reasonable approximation of neural network learning) for different variants of SGD. Constant stepsize SGD (with last iterate or tail-averaging) is investigated in Jain et al. (2018), in Dieuleveut & Bach (2016) in an RKHS frameowrk and also in Mücke et al. (2019) with additional mini-batching, see also Mücke & Reiss (2020) for a more general analysis in Hilbert scales. In Zou et al. (2021b;a) it is shown that benign overfitting also occurs for SGD. Multi-pass SGD is analyzed in Lin et al. (2016); Jain et al. (2016); Lin & Rosasco (2017); Zou et al. (2022) while last iterate bounds can be found in Jain et al. (2019); Wu et al. (2022); Varre et al. (2021).

Despite the attractive statistical properties of all these SGD variants, the complexity of computing regression estimates prevents it from being routinely used in large-scale problems. More precisely, the time complexity and space complexity of SGD and other regularization methods in a standard implementation scale as $\mathcal{O}(n^\alpha)$, $\alpha \in [2,3]$. Such scalings are prohibitive when the sample size $n$ is large.

Distributed learning (DL) based on a divided-and-conquer approach is an effective way to analyze large scale data that can not be handled by a single machine. In this paper we study a distributed learning strategy in linear regression (including both underparameterized and overparameterized regimes) via (tail-) averaged stochastic gradient descent with constant stepsize (DSGD). The approach is quite simple and communication efficient: The training data is distributed across several computing nodes where on each a local SGD is run. In a final step, these local estimates are aggregated (a.k.a. *one-shot SGD*). Local SGD has become state of the art in large scale distributed learning, showing a linear speed-up in the number of workers for convex problems, see e.g. Mcdonald et al. (2009); Zinkevich et al. (2010); Dieuleveut & Patel (2019); Stich (2018); Spiridonoff et al. (2021) and references therein.

The field of DL has gained increasing attention in statistical learning theory with the aim of deriving conditions under which minimax optimal rates of convergence can be guaranteed, see e.g. Chen & Xie (2014), Mackey et al. (2011), Xu et al. (2019), Fan et al. (2019), Shi et al. (2018), Battey et al. (2018), Fan et al. (2021), Bao & Xiong (2021). Indeed, the learning properties of DL in regression settings over Hilbert

spaces are widely well understood. The authors in Zhang et al. (2015) analyze distributed (kernel) ridge regression and show optimal learning rates with appropriate regularization, provided the number of machines increases sufficiently slowly with the sample size, though under restrictive assumptions on the eigenfunctions of the kernel integral operator. This has been alleviated in Lin et al. (2017). However, in these works the number of machines *saturates* if the target is very smooth, meaning that large parallelization seems not possible in this regime.

An extension of these works to more general spectral regularization algorithms for nonparametric least square regression in (reproducing kernel) Hilbert spaces is given in Guo et al. (2017), Mücke & Blanchard (2018), including gradient descent (Lin & Zhou, 2018) and stochastic gradient descent (Lin & Cevher, 2018). The recent work Tong (2021) studies DL for functional linear regression.

We finally mention the work of Mücke et al. (2022), where distributed ordinary least squares (DOLS) in over-parameterized linear regression is studied, i.e. one-shot OLS without any explicit or implicit regularization. It is shown that the number of workers acts as a regularization parameter itself.

**Contributions.** We analyze the performance of DSGD with constant stepsize in overparameterized linear regression and provide upper bounds with matching lower bounds for the excess risk under suitable noise assumptions. Our results show that optimal rates of convergence can be achieved if the number of local nodes grows sufficiently slowly with the sample size. The excess risk as a function of data splits remains constant until a certain threshold is reached. This threshold depends on the structural assumptions imposed on the problem, i.e. on the eigenvalue decay of the Hessian and the coefficients of the true regression parameter.

We additionally perform a comparison between DSGD and DRR, showing that the excess risk of DSGD is upper bounded by the excess risk of DRR under an assumption on the sample complexity (SC) of DSGD, depending on the same structural assumptions. We show that the SC of DSGD remains within constant factors of the SC of DRR.

Our analysis extends known results in this direction from Zou et al. (2021b;a) for the single machine case to the distributed learning setting and from DOLS in Mücke et al. (2022) to SGD with implicit regularization.

**Organization.** In Section 2 we define the mathematical framework needed to present our main results in Section 3, where we provide a theoretical analysis of DSGD with a discussion of our results. In Section 4 we compare DSGD with DRR while Section 5 is devoted to showing some numerical illustrations. The proofs a deferred to the Appendix.

**Notation.** By $\mathcal{L}(\mathcal{H}_1, \mathcal{H}_2)$ we denote the space of bounded linear operators between real Hilbert spaces $\mathcal{H}_1$, $\mathcal{H}_2$. We write $\mathcal{L}(\mathcal{H}, \mathcal{H}) = \mathcal{L}(\mathcal{H})$. For $\mathbf{A} \in \mathcal{L}(\mathcal{H})$ we denote by $\mathbf{A}^T$ the adjoint operator. By $\mathbf{A}^\dagger$ we denote the pseudoinverse of $\mathbf{A}$ and for $w \in \mathcal{H}$ we write $||w||_{\mathbf{A}}^2 := ||\mathbf{A}^{\frac{1}{2}} w||$ for an PSD operator $\mathbf{A}$.

We let $[n] = \{1, ..., n\}$ for every $n \in \mathbb{N}$. For two positive sequences $(a_n)_n, (b_n)_n$ we write $a_n \lesssim b_n$ if $a_n \leq cb_n$ for some $c > 0$ and $a_n \simeq b_n$ if both $a_n \lesssim b_n$ and $b_n \lesssim a_n$.

## 2 SETUP

In this section we provide the mathematical framework for our analysis. More specifically, we introduce distributed SGD and state the main assumptions on our model.

### 2.1 SGD and linear regression

We consider a linear regression model over a real separable Hilbert space $\mathcal{H}$ in random design. More precisely, we are given a random covariate vector $x \in \mathcal{H}$ and a random output $y \in \mathbb{R}$ following the model

$$y = \langle w^*, x \rangle + \epsilon \,, \tag{1}$$

where $\epsilon \in \mathbb{R}$ is a noise variable. We will impose some assumptions on the noise model in Section 3. The true regression parameter $w^* \in \mathcal{H}$ minimizes the least squares test risk, i.e.

$$L(w^*) = \min_{w \in \mathcal{H}} L(w) , \quad L(w) := \frac{1}{2}\mathbb{E}[(y - \langle w, x \rangle)^2] ,$$

where the expectation is taken with respect to the joint distribution $\mathbb{P}$ of the pair $(x, y) \in \mathcal{H} \times \mathbb{R}$. More specifically, we let $w^*$ be the minimum norm element in the set of all minimizers of $L$.

To derive an estimator $\hat{w} \in \mathcal{H}$ for $w^*$ we are given an i.i.d. dataset

$$D := \{(x_1, y_1), ..., (x_n, y_n)\} \subset \mathcal{H} \times \mathbb{R} ,$$

following the above model equation 1, i.e.,

$$\mathbf{Y} = \mathbf{X}w^* + \varepsilon ,$$

with i.i.d. noise $\varepsilon = (\varepsilon_1, ..., \varepsilon_n) \in \mathbb{R}^n$. The corresponding random vector of outputs is denoted as $\mathbf{Y} = (y_1, ..., y_n)^T \in \mathbb{R}^n$ and we arrange the data $x_j \in \mathcal{H}$ into a *data matrix* $\mathbf{X} \in \mathcal{L}(\mathcal{H}, \mathbb{R}^n)$ by setting $(\mathbf{X}v)_j = \langle x_j, v \rangle$ for $v \in \mathcal{H}, 1 \leq j \leq n$. If $\mathcal{H} = \mathbb{R}^d$, then $\mathbf{X}$ is a $n \times d$ matrix (with row vectors $x_j$). We are particular interested in the overparameterized regime, i.e. where $dim(\mathcal{H}) > n$.

In the classical setting of stochastic approximation with constant stepsize, the SGD iterates are computed by the recursion

$$w_{t+1} = w_t - \gamma(\langle w_t, x_t \rangle - y_t)x_t , \quad t = 1, ..., n ,$$

with some initialization $w_1 \in \mathcal{H}$ and where $\gamma > 0$ is the stepsize. The tail average of the iterates is denoted by

$$\bar{w}_{\frac{n}{2}:n} := \frac{1}{n - n/2} \sum_{t=n/2+1}^{n} w_t , \tag{2}$$

and where we denote by $\bar{w}_n := \bar{w}_{0:n}$ the full (uniform) average.

Various forms of SGD (with iterate averaging, tail averaging, multi passes) in the setting of overparameterized linear regression has been analyzed recently in Zou et al. (2021b), Wu et al. (2022), Zou et al. (2022), respectively. In particular, the phenomenon of *benign overfitting* is theoretically investigated in these works. It could be shown that benign overfitting occurs in this setting, i.e. the SGD estimator fits training data very well and still generalizes.

We are interested in this phenomenon for localized SGD, i.e. when our training data is distributed over several computing devices.

## 2.2 Local SGD

In the distributed setting, our data are evenly divided into $M \in \mathbb{N}$ local disjoint subsets

$$D = D_1 \cup ... \cup D_M$$

of size $|D_m| = \frac{n}{M}$, for $m = 1, ..., M$. To each local dataset we associate a *local design matrix* $\mathbf{X}_m \in \mathcal{L}(\mathcal{H}, \mathbb{R}^{\frac{n}{M}})$ (build with local row vectors $x_j^{(m)}$) with local output vector $\mathbf{Y}_m \in \mathbb{R}^{\frac{n}{M}}$ and a local noise vector $\varepsilon_m \in \mathbb{R}^{\frac{n}{M}}$.

The local SGD iterates are defined as

$$w_{t+1}^{(m)} = w_t^{(m)} - \gamma\left(\left\langle w_t^{(m)}, x_t^{(m)} \right\rangle - y_t\right)x_t^{(m)} ,$$

for $t = 1, ..., \frac{n}{M}$ and $m = 1, ..., M$. The averaged local iterates $\bar{w}_{\frac{n}{M}}^{(m)}$ are computed according to equation 2. We are finally interested in the uniform average of the local SGD iterates, building a global estimator:

$$\overline{\overline{w}}_M := \frac{1}{M} \sum_{m=1}^{M} \bar{w}_{\frac{n}{M}}^{(m)} .$$

Distributed learning in overparameterized linear regression is studied in Mücke et al. (2022) for the ordinary least squares estimator (OLS), i.e. without any implicit or explicit regularization and with local interpolation. It is shown that local overfitting is harmless and regularization is done by the number of data splits.

We aim at finding optimal bounds for the excess risk

$$\mathbb{E}\big[L(\overline{\overline{w}}_M)\big] - L(w^*) \,,$$

of distributed SGD (DSGD) with potential local overparameterization and as function of the number of local nodes $M$ and under various model assumptions, to be given in the next section.

## 3 MAIN RESULTS

In this section we present our main results. To do so, we first impose some model assumptions.

**Definition 3.1.**

1. *We define the second moment of $x \sim \mathbb{P}_x$ to be the operator $\mathbf{H} : \mathcal{H} \to \mathcal{H}$, given by*

$$\mathbf{H} := \mathbb{E}[x \otimes x] = \mathbb{E}[\langle \cdot, x \rangle x] \,.$$

2. *The fourth moment operator $\mathbf{M} : \mathcal{L}(\mathcal{H}) \to \mathcal{L}(\mathcal{H})$ is defined by*

$$\mathbf{M} := \mathbb{E}[x \otimes x \otimes x \otimes x] \,,$$

*with $\mathbf{M}(\mathbf{A})(w) = \mathbb{E}[\langle x, \mathbf{A}x \rangle \langle w, x \rangle x]$, for all $w \in \mathcal{H}$.*

3. *The covariance operator of the gradient noise at $w^*$ is defined as $\mathbf{\Sigma} : \mathcal{H} \to \mathcal{H}$,*

$$\mathbf{\Sigma} := \mathbb{E}[(\langle w^*, x \rangle - y)^2 \, x \otimes x] \,.$$

**Assumption 3.2** (Second Moment Condition)**.** *We assume that $\mathbb{E}[y^2|x] < \infty$ almost surely. Moreover, we assume that the trace of $\mathbf{H}$ is finite, i.e., $\mathrm{Tr}[\mathbf{H}] < \infty$.*

**Assumption 3.3** (Fourth Moment Condition)**.** *We assume there exists a positive constant $\tau > 0$ such that for any PSD operator $\mathbf{A}$, we have*

$$\mathbf{M}(\mathbf{A}) \preceq \tau \, \mathrm{Tr}[\mathbf{H}\mathbf{A}]\mathbf{H} \,.$$

Note that this assumption holds if $\mathbf{H}^{-1}x$ is sub-Gaussian, being a standard assumption in least squares regression, see e.g. Bartlett et al. (2020), Zou et al. (2021b), Tsigler & Bartlett (2020).

**Assumption 3.4** (Noise Condition)**.** *Assume that*

$$\sigma^2 := ||\mathbf{H}^{-\frac{1}{2}}\mathbf{\Sigma}\mathbf{H}^{-\frac{1}{2}}|| < \infty \,.$$

This assumption on the noise is standard in the literature about averaged SGD, see e.g. Zou et al. (2021b), Dieuleveut & Bach (2016).

We introduce some further notation involving the second moment operator $\mathbf{H}$: We denote the eigendecomposition as

$$\mathbf{H} = \sum_{j=1}^{\infty} \lambda_j v_j \otimes v_j \,,$$

where the $\lambda_1 \geq \lambda_2 \geq ...$ are the eigenvalues of $\mathbf{H}$ and the $v_j's$ are the corresponding eigenvectors. For $k \geq 1$, we let

$$\mathbf{H}_{0:k} := \sum_{j=1}^{k} \lambda_j v_j \otimes v_j \,, \quad \mathbf{H}_{k:\infty} := \sum_{j=k+1}^{\infty} \lambda_j v_j \otimes v_j \,.$$

Similarly,

$$\mathbf{I}_{0:k} = \sum_{j=1}^{k} v_j \otimes v_j \; , \quad \mathbf{I}_{k:\infty} := \sum_{j=k+1}^{\infty} v_j \otimes v_j \; .$$

A short calculation shows that for all $w \in \mathcal{H}$ we have

$$||w||_{\mathbf{H}_{0:k}^{\dagger}}^2 = \sum_{j=1}^{k} \frac{\langle w, v_j \rangle^2}{\lambda_j} \; , \quad ||w||_{\mathbf{H}_{k:\infty}}^2 = \sum_{j=k+1}^{\infty} \lambda_j \langle w, v_j \rangle^2 \; .$$

We finally set

$$V_k(n, M) := \frac{k}{n} + \gamma^2 \frac{n}{M^2} \sum_{j=k+1}^{\infty} \lambda_j^2 \; . \tag{3}$$

### 3.1 Upper Bound

We now present an upper bound for the averaged local SGD iterates. The proof relies on a bias-variance decomposition and is given in Appendix A.1.

**Theorem 3.5** (DSGD Upper Bound). *Suppose Assumptions 3.2, 3.3 and 3.4 are satisfied and let $\gamma < \frac{1}{\tau \operatorname{Tr}[\mathbf{H}]}$, $w_1 = 0$. The excess risk for the averaged local SGD estimate satisfies*

$$\mathbb{E}\big[L(\overline{\overline{w}}_M)\big] - L(w^*) \; \leq \; 2\mathrm{Bias}(\overline{\overline{w}}_M) \; + \; 2\mathrm{Var}(\overline{\overline{w}}_M) \; ,$$

*where*

$$\mathrm{Bias}(\overline{\overline{w}}_M) \leq \frac{M^2}{\gamma^2 n^2} ||w^*||_{\mathbf{H}_{0:k^*}^{\dagger}}^2 + ||w^*||_{\mathbf{H}_{k^*:\infty}}^2 \; + \; \frac{2\tau M^2 \big(||w^*||_{\mathbf{I}_{0:k^*}}^2 + \gamma \frac{n}{M} ||w^*||_{\mathbf{H}_{k^*:\infty}}^2 \big)}{\gamma n (1 - \gamma \tau \operatorname{Tr}[\mathbf{H}])} \cdot V_{k^*}(n, M)$$

*and*

$$\mathrm{Var}(\overline{\overline{w}}_M) \; \leq \; \frac{\sigma^2}{1 - \gamma \tau \operatorname{Tr}[\mathbf{H}]} \cdot V_{k^*}(n, M) \; ,$$

*with $k^* = \max\{k : \lambda_k \geq \frac{M}{\gamma n}\}$.*

The excess risk is upper bounded in terms of the bias and variance. Both terms crucially depend on the *effective dimension* $k^* = \max\{k : \lambda_k \geq \frac{M}{\gamma n}\}$, dividing the full Hilbert space $\mathcal{H}$ into two parts. On the part associated to the first largest $k^*$ eigenvalues, the bias may decay faster than on the remaining tail part that is associated to the smaller eigenvalues, see Zou et al. (2021b) in the context of single machine SGD, Bartlett et al. (2020); Tsigler & Bartlett (2020), in the context of single machine ridge regression and Mücke et al. (2022) for distributed ordinary least squares.

Our Theorem 3.5 reveals that the excess risk converges to zero if

$$||w^*||_{\mathbf{H}_{k^*:\infty}}^2 \to 0 \; , \quad \frac{2M^2}{\gamma^2 n^2} ||w^*||_{\mathbf{H}_{0:k^*}^{\dagger}}^2 \to 0$$

and $V_{k^*}(n, M) \to 0$ as $n \to \infty$. This requires the eigenvalues of $\mathbf{H}$ to decay sufficiently fast and to choose the number of local nodes $M = M_n$ to be a sequence of $n$. Note that we have to naturally assume $M_n \lesssim n$. In Subsection 3.3 we provide two specific examples of data distributions with specific choices for $(M_n)_{n \in \mathbb{N}}$ such that the above conditions are met, granting not only convergence but also providing explicit rates of convergence.

### 3.2 Lower Bound

Before we state the lower bounds for the excess risk of the DSGD estimator we need to impose some assumptions.

**Assumption 3.6** (Fourth Moment Lower Bound)**.** *We assume there exists a positive constant $\theta > 0$ such that for any PSD operator $\mathbf{A}$, we have*

$$\mathbf{M}(\mathbf{A}) - \mathbf{H}\mathbf{A}\mathbf{H} \succeq \theta \operatorname{Tr}[\mathbf{H}\mathbf{A}]\mathbf{H} \,.$$

**Assumption 3.7** (Well-Specified Noise)**.** *The second moment operator $\mathbf{H}$ is strictly positive definite with $\operatorname{Tr}[\mathbf{H}] < \infty$. Moreover, the noise $\epsilon$ in equation 1 is independent of $x$ and satisfies*

$$\epsilon \sim \mathcal{N}(0, \sigma_{noise}^2) \,.$$

We now come to the main result whose proof can be found in Appendix A.2.

**Theorem 3.8** (DSGD Lower Bound)**.** *Suppose Assumptions 3.6 and 3.7 are satisfied. Assume $w_1 = 0$. The excess risk of the DSGD estimator satisfies*

$$\mathbb{E}\big[L(\overline{\overline{w}}_M)\big] - L(w^*) \geq \frac{M(M-1)}{100\gamma^2 n^2}\left(||w^*||^2_{\mathbf{H}^\dagger_{0:k^*}} + \frac{\gamma^2 n^2}{M^2}||w^*||^2_{\mathbf{H}_{k^*:\infty}}\right) + \frac{\sigma_{noise}^2}{100} \cdot V_{k^*}(n, M) \,,$$

*where $V_{k^*}(n, M)$ is defined in equation 3.*

The lower bound for the excess risk also decomposes into a bias part (first term) and a part associated to the variance (second term). Comparing the bias with the upper bound for the bias from Theorem 3.5 shows that both are of the same order. Comparing the variances reveals that they are of the same order if

$$\frac{2\tau M^2\big(||w^*||^2_{\mathbf{I}_{0:k^*}} + \gamma\frac{n}{M}||w^*||^2_{\mathbf{H}_{k^*:\infty}}\big)}{\gamma n(1 - \gamma\tau\operatorname{Tr}[\mathbf{H}])} \lesssim 1 \,.$$

In the next section, we will provide specific conditions and examples when this is satisfied.

### 3.3 Fast Rates of convergence for specific distributions

We now consider two particular cases of data distributions, namely the *spiked covariance model* (with local overparameterization) and the case where the eigenvalues of the second moment operator $\mathbf{H}$ decay *polynomially*. These are standard assumptions for the model, see e.g. Tsigler & Bartlett (2020); Zou et al. (2021b); Mücke et al. (2022). In both cases, we determine a range of the number of local nodes $M_n$ depending on the global sample size such that the bias is dominated by the variance. The final error is then of the order of the variance, if the number of local nodes grows sufficiently slowly with the sample size. The optimal[1] number exactly balances bias and variance.

**Corollary 3.9** (Spiked Covariance Model)**.** *Suppose all assumptions of Theorem 3.5 are satisfied. Assume that $||w^*|| \leq R$ for some $R > 0$ and $\mathbf{H} \in \mathbb{R}^{d \times d}$. Let $d = \left(\frac{n}{M}\right)^q$ for some $q > 1$ and $\tilde{d} = \left(\frac{n}{M}\right)^r < d$ for some $0 < r \leq 1$. Suppose the spectrum of $\mathbf{H}$ satisfies*

$$\lambda_j = \begin{cases} \frac{1}{\tilde{d}} & : j \leq \tilde{d} \\ \frac{1}{d - \tilde{d}} & : \tilde{d} + 1 \leq j \leq d \,. \end{cases}$$

*If*

$$M_n \leq \sqrt{\frac{\gamma(1 - 2\gamma\tau)n}{R^2}}$$

---

[1] *Optimal* in the sense of the maximal possible number of local nodes that balances bias and variance.

*then for any $n$ sufficiently large, we have*

$$\mathbb{E}\big[L(\overline{\overline{w}}_{M_n})\big] - L(w^*) \;\leq\; c\,\frac{1}{\gamma M_n}\left(\frac{M_n}{n}\right)^\nu ,$$

*where $\nu = \min\{1 - r, q - 1\}$ and for some $c < \infty$, depending on $\tau, \gamma, \sigma$.*

Choosing the maximum number of local nodes $M_n \simeq \sqrt{n}$ gives the fast rate of order

$$\mathbb{E}\big[L(\overline{\overline{w}}_{M_n})\big] - L(w^*) \;\lesssim\; \left(\frac{1}{n}\right)^{\frac{\nu+1}{2}} .$$

for the excess risk.

**Corollary 3.10** (Polynomial Decay)**.** *Suppose all assumptions of Theorem 3.5 are satisfied with $\gamma <$ $\min\left\{1, \frac{1}{\tau\,\mathrm{Tr}[\mathbf{H}]}\right\}$. Assume that $\|w^*\| \leq R$ for some $R > 0$. Suppose the spectrum[2] of $\mathbf{H}$ satisfies for some $r > 0$*

$$\lambda_j = j^{-(1+r)} .$$

*If*

$$M_n \leq \left(\frac{\gamma}{R^2}\right)^{\frac{1+r}{2+r}} \cdot (\gamma n)^{\frac{1}{2+r}} ,$$

*then for any $n$ sufficiently large, we have*

$$\mathbb{E}\big[(\overline{\overline{w}}_M)\big] - L(w^*) \;\leq\; c\,\frac{\gamma}{M_n}\left(\frac{M_n}{n}\right)^{\frac{r}{1+r}} ,$$

*for some $c < \infty$, depending on $\tau, \gamma, \sigma$.*

Choosing the maximum number of local nodes $M_n \simeq n^{\frac{1}{2+r}}$ gives the fast rate of order

$$\mathbb{E}\big[L(\overline{\overline{w}}_{M_n})\big] - L(w^*) \;\lesssim\; \left(\frac{1}{n}\right)^{\frac{r+1}{r+2}} .$$

for the excess risk.

### 3.4 Discussion

**Comparison to single machine SGD.** We compare the DSGD algorithm with the single machine SGD algorithm, i.e. when $M = 1$. For this case, we recover the results from Zou et al. (2021b) under the same assumptions. Our Corollaries 3.9, 3.10 show that the excess risk is dominated by the variance as long as $M$ grows sufficiently slowly with the sample size. But we can say even more: In the spiked covariance model, if $M_n \simeq n^\beta$ for $\beta \in [0, 1/2]$, we see that DSGD performs as good as single machine SGD, provided $\nu \leq 1$. Indeed, a direct comparison shows that

$$\frac{1}{\gamma M_n}\left(\frac{M_n}{n}\right)^\nu \simeq \frac{1}{\gamma n^\beta}\left(\frac{n^\beta}{n}\right)^\nu \simeq \frac{1}{\gamma}\left(\frac{1}{n}\right)^\nu ,$$

for any $\beta \in [0, 1/2]$ and $\nu \leq 1$. Recall that all our bounds are of optimal order, hence the relative efficiency remains of constant order until the critical threshold for $M_n$ is reached.
However, if $M_n$ is larger than the threshold, i.e. if $\beta \in (1/2, 1]$, then the bias term is dominating. In this case, the excess risk is of order

$$\frac{2M^2}{n^2}\|w^*\|^2_{\mathbf{H}_{0:k^*}^\dagger} + \|w^*\|^2_{\mathbf{H}_{k^*:\infty}} \simeq \left(\frac{M_n}{n}\right)^{2-r} + \left(\frac{M_n}{n}\right)^q \simeq \left(\frac{n^\beta}{n}\right)^{2-r} + \left(\frac{n^\beta}{n}\right)^q ,$$

---

[2]Note that the choice $\lambda_j = j^{-(1+r)}$ ensures that $\mathrm{Tr}[\mathbf{H}] < \infty$.

being larger than the variance, see the proof of Corollary 3.9, Appendix A.3.

The same observations can be made for the setting in Corollary 3.10 when the eigenvalues are polynomially decaying. If we let $M_n \simeq n^\beta$ with $\beta \in [0, 1/(2+r)]$, then the variance dominates and for all $r > 0$, the test error satisfies

$$\frac{1}{\gamma M_n} \left( \frac{M_n}{n} \right)^{\frac{r}{r+1}} \simeq \frac{1}{\gamma n^\beta} \left( \frac{n^\beta}{n} \right)^{\frac{r}{r+1}} \simeq \frac{1}{\gamma} \left( \frac{1}{n} \right)^{\frac{r}{r+1}}.$$

We refer to Section5 and Section C for some numerical experiments.

**Comparison to distributed learning in RKHSs.** We emphasize that all our results above hold for a constant stepsize $0 < \gamma < \min\left\{ 1, \frac{1}{\tau \text{Tr}[\mathbf{H}]} \right\}$. In particular, $\gamma$ does not depend on the number $M$ of local nodes. This result is line with the results for regularized distributed learning over reproducing kernel Hilbert spaces, see Zhang et al. (2015); Lin et al. (2017); Mücke & Blanchard (2018) and references therein. In this setting it is shown for a large class of spectral regularization methods[3] that the optimal regularization parameter $\lambda$ that leads to minimax optimal bounds, depends on the global sample size only and is of order $n^{-\alpha}$, $\alpha \in (0, 1]$. In particular, this parameter is chosen as in the single machine machine setting and each local subproblem is underregularized. This leads to a roughly constant bias (unchanged by averaging) in the distributed setting, an increase in variance but averaging reduces the variance sufficiently to obtain optimal excess risk bounds. The same phenomenon occurs in our DSGD setting. On each local node the same stepsize $\gamma$ as for the $M = 1$ case is applied.

**Comparison to distributed ordinary least squares (DOLS).** We also compare our results with those recently obtained in Mücke et al. (2022) for DOLS in random design linear regression. The general observation in this work is that in the presence of overparameterization, the number of local nodes acts as a regularization parameter, balancing bias and variance. Recall that this is in contrast to what we observe for DSGD due to the implicit regularization. The optimal number of splits $M_{opt}^{OLS}$ depends on structural assumptions, i.e. eigenvalue decay and decay of the Fourier coefficients of $w^*$ (a.k.a. *source condition*).

For the spiked covariance model, the optimal number $M_n^{OLS}$ of DOLS is of order

$$M_n^{OLS} \simeq \left( \frac{d n^{3/2}}{d \cdot \tilde{d}} \right)^{2/5} \simeq n^{\frac{3-2r}{5-2r}},$$

see Corollary 3.14 in Mücke et al. (2022). Comparing with our maximum number for $M_n \simeq n^{1/2}$ from our Corollary 3.9 we observe that $M_n^{OLS} \lesssim M_n^{SGD}$ if $\frac{1}{2} \leq r \leq 1$, i.e., DSGD allows for more parallelization in this regime.

For polynomially decaying eigenvalues $\lambda_j \sim j^{1+r}$, $r > 0$, the optimal number of data splits in Corollary 3.9 (Mücke et al., 2022) scales as $M_n^{OLS} \simeq n^{1/3}$. Compared to our result from Corollary 3.10 we have

$$M_n^{SGD} \simeq n^{\frac{1}{2+r}} \lesssim n^{1/3}$$

for all $r \geq 1$. Thus, DOLS seems to allow more data splits under optimality guarantees for fast polynomial decay, i.e. large $r$.

## 4 COMPARISON OF SAMPLE COMPLEXITY OF DSGD AND DRR

In this section we compare the distributed tail-averaged SGD estimator with the distributed Ridge Regression (RR) estimator (see Zhang et al. (2015); Lin et al. (2017); Mücke & Blanchard (2018); Sheng & Dobriban (2020) or Tsigler & Bartlett (2020) for RR in the single machine case). Recall that RR reduces to ordinary least-squares (OLS) if the regularization parameter is set to zero. As a special case, we compare our results to local OLS from Mücke et al. (2022) and analyze the benefit of implicit regularization of local SGD in the presence of local overparameterization.

---

[3]This class contains, among others, gradient descent and accelerated methods like Heavy ball and Nesterov, ridge regression or PCA.

We recall that for any $m \in [M]$, $\lambda \geq 0$, the local RR estimates are defined by

$$\hat{w}_m^{\text{RR}}(\lambda) = \mathbf{X}_m^T (\mathbf{X}_m \mathbf{X}_m^T + \lambda)^{-1} \mathbf{Y}_m .$$

The average is

$$\overline{w}_n^{\text{RR}}(\lambda) = \frac{1}{M} \sum_{m=1}^{M} \hat{w}_m^{\text{RR}} .$$

We aim at showing that the excess risk of DSGD is upper bounded by the excess risk of DRR under suitable assumptions on the sample complexity. To this end, we first derive a lower bound for DRR to compare with. The proof follows by combining Proposition B.3 and Proposition B.5 with Lemma B.2.

**Assumption 4.1.** *The variable $\mathbf{H}^{-1} x$ is sub-Gaussian and has independent components.*

Similarly to the bounds for DSGD, our bounds for DRR depend on the effective dimension

$$k_{\text{RR}}^* := \min \left\{ k : \lambda_{k+1} \leq \frac{M\left(\lambda + \sum_{j>k} \lambda_j\right)}{bn} \right\} ,$$

for $\lambda > 0$ and some $b > 1$.

**Theorem 4.2** (Lower Bound Distributed RR). *Suppose Assumption 4.1 holds and that $\mathbf{H}$ is strictly positive definite with $\text{Tr}[\mathbf{H}] < \infty$. Assume that $k_{\text{RR}}^* \leq \frac{n}{c'M}$ for some $c' > 1$. There exist constants $b, c > 1$ such that the excess risk of the averaged RR estimator satisfies*

$$\mathbb{E}\left[L(\overline{w}_n^{\text{RR}}(\lambda))\right] - L(w^*) \geq \|w^*\|_{\mathbf{H}_{k_{\text{RR}}^*:\infty}}^2 + \frac{M^2 \left(\lambda + \sum_{j>k_{\text{RR}}^*} \lambda_j\right)^2}{cn^2} \cdot \|w^*\|_{\mathbf{H}_{0:k_{\text{RR}}^*}^{-1}}^2$$
$$+ \frac{\sigma^2}{c} \left( \frac{k_{\text{RR}}^*}{n} + \frac{n}{M^2} \cdot \frac{\sum_{j>k_{\text{RR}}^*} \lambda_j^2}{(\lambda + \sum_{j>k_{\text{RR}}^*} \lambda_j)^2} \right) .$$

We do our risk comparison particularly for tail-averaged DSGD and derive a bias-improved upper bound. The proof is given in Section B.2 and is an extension of Lemma 6.1 in Zou et al. (2021a) to DSGD.

**Theorem 4.3** (Upper Bound Tail-averaged DSGD). *Suppose Assumption 3.7 is satisfied. Let $\overline{\overline{w}}_{M_n}$ denote the tail-averaged distributed estimator with $n$ training samples and assume $\gamma < 1/\text{Tr}[H]$. For arbitrary $k_1, k_2 \in [d]$*

$$\mathbb{E}\left[L(\overline{\overline{w}}_M)\right] - L(w^*) = \text{Bias}(\overline{\overline{w}}_M) + \text{Var}(\overline{\overline{w}}_M)$$

*with*

$$\text{Bias}(\overline{\overline{w}}_M) \leq \frac{c_b M^2}{\gamma^2 n^2} \cdot \left\| \exp\left(-\frac{n}{M} \gamma \mathbf{H}\right) w^* \right\|_{\mathbf{H}_{0:k_1}^{-1}}^2 + \|w^*\|_{\mathbf{H}_{k_1:\infty}}^2 ,$$

$$\text{Var}(\overline{\overline{w}}_M) \leq c_v (1 + R^2) \cdot \sigma^2 \left( \frac{k_2}{n} + \frac{n\gamma^2}{M^2} \cdot \sum_{j>k_2} \lambda_j^2 \right) ,$$

*for some universal constants $c_b, c_v > 0$.*

To derive the risk comparison we fix a sample size $n_{\text{RR}}$ and $n_{\text{SGD}}$ for DRR and tail-averaged DSGD, resp., and derive conditions on the sample complexities such that individually, the bias and variance of DSGD is upper bounded by the bias and variance of DRR, respectively. Combining then both of the above theorems finally leads to the risk comparison result. A detailed computation is given in Section B.3.

**Theorem 4.4** (Comparison DSGD with DRR). *Let $\overline{\overline{w}}_{M_{n_{\mathrm{SGD}}}}$ denote the tail-averaged distributed estimator with $n_{\mathrm{SGD}}$ training samples. Let further $\overline{w}_{n_{\mathrm{RR}}}^{\mathrm{RR}}(\lambda)$ denote the distributed RR estimator with $n_{\mathrm{RR}}$ training samples and with regularization parameter $\lambda \geq 0$. Suppose all assumptions from Theorems 4.2 ,4.3 are satisfied. There exist constants $b, c > 1$ and $0 < L_{\lambda,\gamma} \leq L'_{\lambda,\gamma}$ such that for $C^* := c\left(1 + \frac{||w^*||^2}{\sigma^2}\right)$,*

$$C_\lambda^* := \lambda + \sum_{j > k_{\mathrm{RR}}^*} \lambda_j \,, \quad \gamma < \min\left\{\frac{1}{\mathrm{Tr}[H]}, \frac{1}{\sqrt{c}C^*C_\lambda^*}\right\} \tag{4}$$

*and*

$$L_{\lambda,\gamma} \cdot n_{\mathrm{RR}} \leq n_{\mathrm{SGD}} \leq L'_{\lambda,\gamma} \cdot n_{\mathrm{RR}}$$

*the excess risks of DSGD and DRR satisfy*

$$\mathbb{E}\left[L(\overline{\overline{w}}_M)\right] - L(w^*) \ \leq \ \mathbb{E}\left[L(\overline{w}_{n_{\mathrm{RR}}}^{\mathrm{RR}}(\lambda))\right] - L(w^*) \,. \tag{5}$$

*The constants $L_{\lambda,\gamma}, L'_{\lambda,\gamma}$ are explicitly given by*

$$L_{\lambda,\gamma} = \max\left\{C^*, \frac{\sqrt{c(1 - \gamma\lambda_{k_{\mathrm{RR}}^*})}}{\gamma C_\lambda^*}\right\} \,, \quad L'_{\lambda,\gamma} = \frac{1}{C^*\gamma^2(C_\lambda^*)^2} \,.$$

Note that in the above Theorem, assumption equation 4 on the stepsize ensures that $0 < L_{\lambda,\gamma} \leq L'_{\lambda,\gamma}$.
Our result shows that DSGD performs better than DRR/ DOLS if the sample complexity (SC) of SGD differs from the SC of RR/OLS by no more than a constant. This constant depends on the amount of regularization $\lambda$, the stepsize $\gamma$ and the tail behavior of the eigenvalues of the Hessian. We refer to Section B.3.2 for a more detailed discussion. Our bound slightly differs from Zou et al. (2021a) for the case $M = 1$ in two respects: We scale our SC such that the constant in equation 5 is equal to one while Zou et al. (2021a) show that both risks are of the same order (with a constant larger than one). Second, we also show that the SC of DSGD is upper bounded by a factor of the SC of DRR/DOLS while Zou et al. (2021a) only derive a lower bound. However, we remark that $n_{\mathrm{SGD}}$ in our Theorem is larger that $n_{\mathrm{RR}}$ as the constant $L_{\lambda,\gamma} \geq 1$. A look onto Figure 1 reveals that optimally tuned DSGD may perform better than optimally tuned DRR even with the same or smaller sample size for certain problem instances. This suggests that our bound may be refined.

## 5 NUMERICAL EXPERIMENTS

We illustrate our theoretical findings with experiments on simulated and real data. The reader may find additional experiments in Section C.

**Simulated Data.** In a first experiment in Figure 1 (left) we analyze the test error of DSGD as a function of the local nodes $M$. We generate $n = 500$ i.i.d. training data with $x_j \sim \mathcal{N}(0, \mathbf{H})$ with mildly overparameterization $d = 700$. The target $w^*$ satisfies three different decay conditions $w_j^* = j^{-\alpha}$, $\alpha \in \{0, 1, 10\}$. The eigenvalues of $\mathbf{H}$ follow a polynomial decay $\lambda_j = j^{-2}$. The local nodes satisfy $M_n = n^\beta$, $\beta \in \{0, 0.1, ..., 0.9\}$. According to Corollary 3.10 we see that a fast decay of $w_j^*$ (i.e. a smaller norm $||w^*||$) allows for more parallelization until the test error blows up.
In a second experiment we compare the sample complexity of optimally tuned tail-averaged DSGD and DRR for different sources $w^*$, see Figures 1 (right), 2. Here, the data are generated as above with $d = 200$, $\lambda_j = j^{-2}$ and $w_j^* = j^{-\alpha}$, $\alpha \in \{0, 1, 10\}$. The number of local nodes is fixed at $M_n = n^{1/3}$ for each $n \in \{100, ..., 6000\}$. For this problem instance, DSGD may perform even better than DRR for sparse targets ($\alpha = 10$), i.e., DSGD achieves the same accuracy as DRR with less samples in this regime. For less sparse targets $\alpha = 1$, the sample complexities of DSGD and DRR are comparable while for non-sparse targets ($\alpha = 0$), DRR outperforms DSGD.

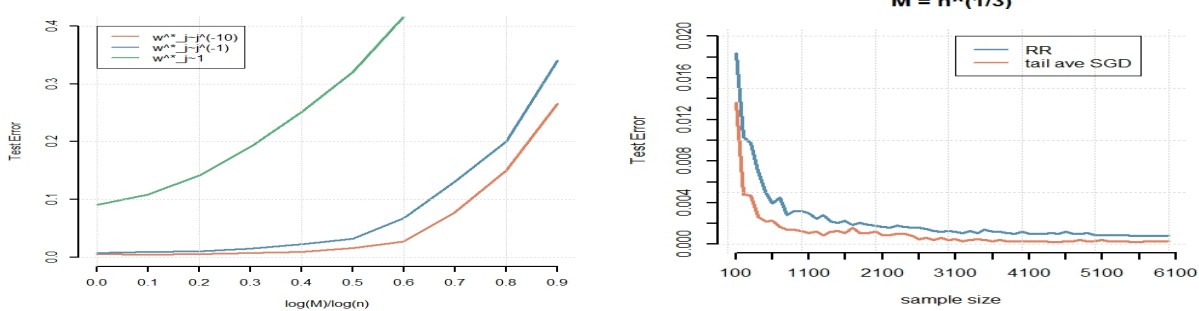

Figure 1: **Left:** Test error for DSGD with $\lambda_j = j^{-2}$ for different sources $w^*$ as a function of $M$.
**Right:** Comparison of optimally tuned tail-ave DSGD with DRR with $\lambda_j = j^{-2}$, $w_j^* = j^{-10}$, $M_n = n^{1/3}$.

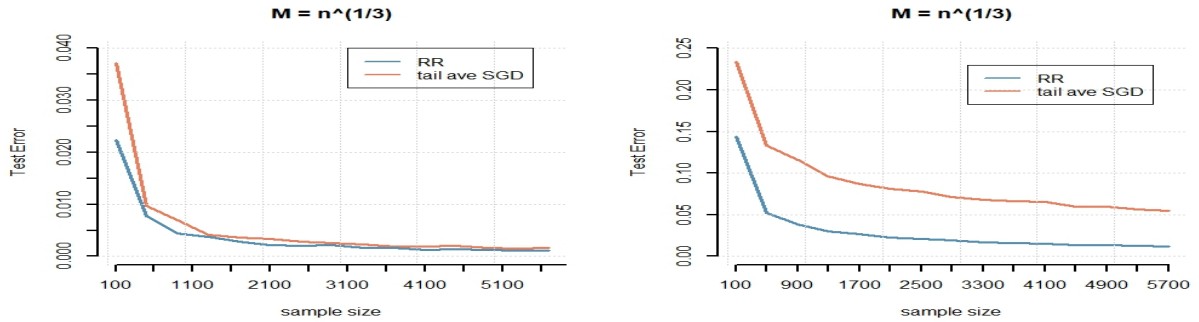

Figure 2: Comparison of optimally tuned tail-ave DSGD with DRR with $\lambda_j = j^{-2}$ for different sources $w^*$, with $\lambda_j = j^{-2}$ and $M_n = n^{1/3}$. **Left:** $w_j^* = j^{-1}$ **Right:** $w_j^* = 1$.

**Real Data.** To analyze the performance of DSGD on real data, we considered the classification problem of the Gisette data set[4], containing pictures of the digits four and nine. We used the first 3000 samples of the original train data set for training and the second 3000 samples for evaluation. The feature dimension of one picture is $d = 5000$. Hyper-parameters had been fine-tuned on the validation data set to achieve the best performance. The first experiment in Figure 3(left) again analyzes the test error of DSGD as a function of the local nodes $M$. Because the feature dimension is quite large, the optimal stepsize is small ($\gamma \sim 10^{-10}$). Theorem 3.8 therefore explains why in our example the bias-term and thus the test error grows rather quickly with the number of local nodes. In Figure 3(right) we compare DRR with tail- and full-averaged DSGD. We observe that DRR slightly outperforms DSG. According to Theorem 4.4, we need sparsity for $w^*$ so that DSGD can keep up with DRR. This might be not the case for the Gisette data set.

# 6 Summary

We analyzed the performance of distributed constant stepsize (tail-) averaged SGD for linear regression in an overparameterized regime. We find that the relative efficiency as a function of the number of workers remains largely unchanged until a certain threshold is reached. This threshold depends on the structural assumptions imposed by the problem at hand (eigenvalue decay of the Hessian $\mathbf{H}$ and the norm of the target $w^*$). This is in contrast to distributed OLS without any implicit or explicit regularization with local overparameterization, where the number of workers itself acts as a regularization parameter, see Figure 4 in Appendix C.
We also compared the sample complexity of DSGD and DRR and find that the sample complexity of DSGD remains within constant factors of the sample complexity of DRR. For some problem instances, tail-averaged

---

[4]http://archive.ics.uci.edu/ml/datasets/Gisette

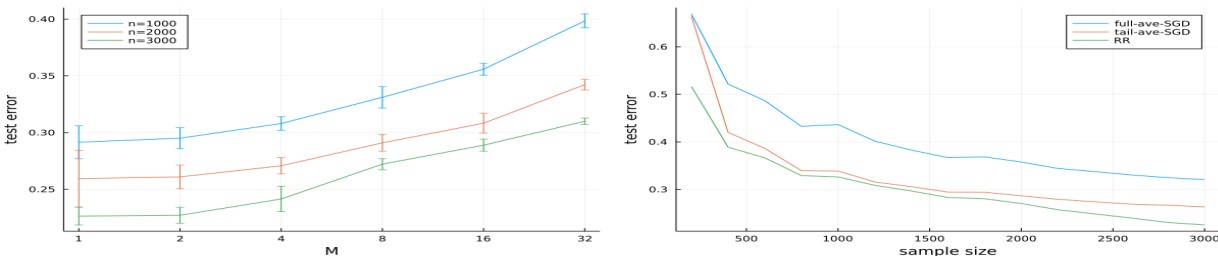

Figure 3: **Left:** Test error for DSGD with $n = 1000, 2000, 3000$ and different $M$. **Right:** Comparison of DSGD with DRR for $M_n = n^{1/4}$.

SGD may outperform DRR, i.e., achieves the same or better accuracy with less samples. Our bound is not sharp and may be improved in future research.

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

**Notation.** By $\mathcal{L}(\mathcal{H}_1, \mathcal{H}_2)$ we denote the space of bounded linear operators between real Hilbert spaces $\mathcal{H}_1$, $\mathcal{H}_2$ with operator norm $|| \cdot ||$. We write $\mathcal{L}(\mathcal{H}, \mathcal{H}) = \mathcal{L}(\mathcal{H})$. For $\mathbf{A} \in \mathcal{L}(\mathcal{H})$ we denote by $\mathbf{A}^T$ the adjoint operator. For two PSD operators on $\mathcal{H}$ we write $\mathbf{A} \preceq \mathbf{B}$ if $\langle (\mathbf{A} - \mathbf{B})v, v \rangle \geq 0$ for all $v \in \mathcal{H}$. We further let $\langle \mathbf{A}, \mathbf{B} \rangle_{op} = \text{Tr}[\mathbf{A}^T \mathbf{B}]$.

# A  PROOFS SECTION 3 (BOUNDS FOR DSGD)

## A.1  Proofs Upper Bound

### A.1.1  Bias-Variance Decomposition

We will use an iterative bias-variance-decomposition which has been extensively studied before in the non distributed case (see Jain et al. (2016), Zou et al. (2021b)). First we need a couple of definitions.

-) **Centered local iterates:** Set $\eta_t^{(m)} := w_t^{(m)} - w^*$ and

$$\bar{\eta}_n^{(m)} := \frac{M}{n} \sum_{t=1}^{n/M} \eta_t^{(m)} \; , \quad \bar{\bar{\eta}}_M := \frac{1}{M} \sum_{m=1}^{M} \bar{\eta}_n^{(m)} .$$

-) **Local bias:** For $m = 1, ..., M$ we set $b_1^{(m)} = w_1 - w^*$,

$$b_t^{(m)} := (\mathbf{I} - \gamma x_t^{(m)} \otimes x_t^{(m)}) b_{t-1}^{(m)} \; , \quad t = 2, ..., \frac{n}{M}$$

$$\bar{b}_n^{(m)} := \frac{M}{n} \sum_{t=1}^{n/M} b_t^{(m)} \; , \quad \bar{\bar{b}}_M := \frac{1}{M} \sum_{m=1}^{M} \bar{b}_n^{(m)} .$$

-) **Local variance:** For $m = 1, ..., M$ we set $v_1^{(m)} = 0$ and

$$v_t^{(m)} := (\mathbf{I} - \gamma x_t^{(m)} \otimes x_t^{(m)}) v_{t-1}^{(m)} + \gamma \epsilon_t^{(m)} x_t^{(m)} \; , \quad t = 2, ..., \frac{n}{M} \; ,$$

$$\bar{v}_n^{(m)} := \frac{M}{n} \sum_{t=1}^{n/M} v_t^{(m)} \; , \quad \bar{\bar{v}}_M := \frac{1}{M} \sum_{m=1}^{M} \bar{v}_n^{(m)} ,$$

where we let $\epsilon_t^{(m)} := y_t^{(m)} - \left\langle x_t^{(m)}, w^* \right\rangle$.

Note that for any $m = 1, ..., M$ and $t \geq 1$ one has

$$\mathbb{E}[b_{t+1}^{(m)}] = \mathbb{E}[\mathbb{E}[b_{t+1}^{(m)} | b_t^{(m)}]] = \mathbb{E}[\mathbb{E}[(\mathbf{I} - \gamma x_{t+1}^{(m)} \otimes x_{t+1}^{(m)}) b_t^{(m)} | b_t^{(m)}]] = (\mathbf{I} - \gamma \mathbf{H}) \mathbb{E}[b_t^{(m)}] . \tag{6}$$

Moreover, from B.4 in Zou et al. (2021b), we find

$$\mathbb{E}[v_{t+1}^{(m)}] = (\mathbf{I} - \gamma H) \mathbb{E}[v_t^{(m)}] = (\mathbf{I} - \gamma H)^t \mathbb{E}[v_1^{(m)}] = 0 . \tag{7}$$

It is easy to see that $\eta_t^{(m)} = b_t^{(m)} + v_t^{(m)}$ and therefore

$$\bar{\bar{\eta}}_M = \bar{\bar{b}}_M + \bar{\bar{v}}_M. \tag{8}$$

**Lemma A.1.** *Define*

$$\text{Bias}(\overline{\overline{w}}_M) := \frac{1}{2}\left\langle \mathbf{H}, \mathbb{E}\left[\overline{\overline{b}}_M \otimes \overline{\overline{b}}_M\right]\right\rangle_{op}, \quad \text{Var}(\overline{\overline{w}}_M) := \frac{1}{2}\left\langle \mathbf{H}, \mathbb{E}\left[\overline{\overline{v}}_M \otimes \overline{\overline{v}}_M\right]\right\rangle_{op}.$$

a) *We have the following decomposition for the excess risk,*

$$\mathbb{E}\left[L(\overline{\overline{w}}_M)\right] - L(w^*) \leq \left(\sqrt{\text{Bias}(\overline{\overline{w}}_M)} + \sqrt{\text{Var}(\overline{\overline{w}}_M)}\right)^2.$$

b) *Suppose the model noise $\epsilon_t^{(m)}$ is well-specified, i.e., $\epsilon_t^{(m)} := y_t^{(m)} - \left\langle x_t^{(m)}, w^*\right\rangle$ and $x_t^{(m)}$ are independent and $\mathbb{E}[\epsilon_t^{(m)}] = 0$, then we have the following equality for the excess risk,*

$$\mathbb{E}\left[L(\overline{\overline{w}}_M)\right] - L(w^*) = \text{Bias}(\overline{\overline{w}}_M) + \text{Var}(\overline{\overline{w}}_M).$$

*Proof of Lemma A.1.* The proof strategy is similar to the non distributed case (see Zou et al. (2021b), Lemma B2 and Lemma C1). For completeness we included it here.
a) By definition of the excess risk we have

$$\begin{aligned}
L(\overline{\overline{w}}_M) - L(w^*) &= \frac{1}{2}\int_{\mathcal{H}}\langle\overline{\overline{w}}_M - w^*, x\rangle^2\,\mathbb{P}_{\mathbf{x}}(d\mathbf{x}) \\
&= \frac{1}{2}\langle\mathbf{H}(\overline{\overline{w}}_M - w^*), \overline{\overline{w}}_M - w^*\rangle \\
&= \frac{1}{2}\|\mathbf{H}^{\frac{1}{2}}(\overline{\overline{w}}_M - w^*)\|^2 \\
&= \frac{1}{2}\left\|\overline{\overline{b}} + \overline{\overline{v}}\right\|_{\mathbf{H}}^2,
\end{aligned}$$

where we used (8) for the last equality. Using Cauchy-Schwarz inequality we obtain

$$\begin{aligned}
\mathbb{E}[L(\overline{\overline{w}}_M) - L(w^*)] &\leq \left(\sqrt{\frac{1}{2}\mathbb{E}\left\|\overline{\overline{b}}\right\|_{\mathbf{H}}^2} + \sqrt{\frac{1}{2}\mathbb{E}\|\overline{\overline{v}}\|_{\mathbf{H}}^2}\right)^2 \\
&= \left(\sqrt{\frac{1}{2}\left\langle\mathbf{H}, \mathbb{E}\left[\overline{\overline{b}}_M \otimes \overline{\overline{b}}_M\right]\right\rangle_{op}} + \sqrt{\frac{1}{2}\left\langle\mathbf{H}, \mathbb{E}\left[\overline{\overline{v}}_M \otimes \overline{\overline{v}}_M\right]\right\rangle_{op}}\right)^2
\end{aligned}$$

b) Set $P_t^{(m)} = \mathbf{I} - \gamma x_t^{(m)} \otimes x_t^{(m)}$. Note that we have

$$b_t^{(m)} = \prod_{k=1}^{t} P_k^{(m)} b_0^{(m)}, \qquad\qquad v_t^{(m)} = \gamma\sum_{i=1}^{t}\prod_{j=i+1}^{t}\epsilon_i^{(m)} P_j^{(m)} x_i^{(m)}.$$

By assumption, we therefore have for all $s, t \leq n/M$ and $m, m' \leq M$,

$$\begin{aligned}
\mathbb{E}\left[b_s^{(m)} \otimes v_t^{(m')}\right] &= \gamma\mathbb{E}\left[\prod_{k=1}^{s}P_k^{(m)}b_0^{(m)} \otimes \sum_{i=1}^{t}\prod_{j=i+1}^{t}\epsilon_i^{(m')}P_j^{(m')}x_i^{(m')}\right] \\
&= \gamma\sum_{i=1}^{t}\mathbb{E}\left[\prod_{k=1}^{s}P_k^{(m)}b_0^{(m)} \otimes \prod_{j=i+1}^{t}P_j^{(m')}x_i^{(m')}\right]\mathbb{E}[\epsilon_i^{(m')}] = 0.
\end{aligned}$$

This implies

$$\mathbb{E}\left[\overline{\overline{b}}_M \otimes \overline{\overline{v}}_M\right] = 0. \tag{9}$$

From (8) we therefore have

$$\mathbb{E}\left[\bar{\bar{\eta}}_M \otimes \bar{\bar{\eta}}_M\right] = \mathbb{E}\left[\bar{\bar{b}}_M \otimes \bar{\bar{b}}_M\right] + \mathbb{E}\left[\bar{\bar{v}}_M \otimes \bar{\bar{v}}_M\right] \tag{10}$$

Finally, by definition of the excess risk we have

$$\begin{aligned}
\mathbb{E}[L(\overline{\overline{w}}_M) - L(w^*)] &= \frac{1}{2}\mathbb{E}\left[\int_{\mathcal{H}}\langle\overline{\overline{w}}_M - w^*, x\rangle^2\, \mathbb{P}_{\mathbf{x}}(d\mathbf{x})\right] \\
&= \frac{1}{2}\mathbb{E}\left[\langle\mathbf{H}(\overline{\overline{w}}_M - w^*), \overline{\overline{w}}_M - w^*\rangle\right] \\
&= \frac{1}{2}\left\langle\mathbf{H}, \mathbb{E}\left[\bar{\bar{\eta}}_M \otimes \bar{\bar{\eta}}_M\right]\right\rangle_{op} \tag{11} \\
&= \mathrm{Bias}(\overline{\overline{w}}_M) + \mathrm{Var}(\overline{\overline{w}}_M), \tag{12}
\end{aligned}$$

where we used (10) for the last equality.

$\square$

### A.1.2 Upper Bound

For the non distributed case Zou et al. (2021b) (see Lemma B.11 and Lemma B.6 ) already established upper bounds. More precisely we have for the local bias and variance term:

**Proposition A.2.** *Set $k^* = \max\left\{k : \lambda_k \geq \frac{M}{n\gamma}\right\}$. If the step size satisfies $\gamma < 1/(\tau\,\mathrm{tr}(\mathbf{H}))$, we have for every $m \in [M]$:*

*a) Under Assumption 3.2 and 3.3, it holds that*

$$\begin{aligned}
Bias\left(\bar{w}_{\frac{n}{M}}^{(m)}\right) :=&\frac{1}{2}\left\langle\mathbf{H}, \mathbb{E}\left[b_t^{(m)} \otimes b_t^{(m)}\right]\right\rangle_{op} \\
\leq&\frac{M^2}{\gamma^2 n^2} \cdot \|\mathbf{w}_0 - \mathbf{w}^*\|_{\mathbf{H}_{0:k^*}^{-1}}^2 + \|\mathbf{w}_0 - \mathbf{w}^*\|_{\mathbf{H}_{k^*:\infty}^2} \\
&+ \frac{2\tau M^2\left(\|\mathbf{w}_0 - \mathbf{w}^*\|_{\mathbf{I}_{0:k^*}}^2 + \frac{n}{M}\gamma\|\mathbf{w}_0 - \mathbf{w}^*\|_{\mathbf{H}_{k^*:\infty}}^2\right)}{\gamma n(1 - \gamma\tau\,\mathrm{tr}(\mathbf{H}))} \cdot \left(\frac{k^*}{n} + \frac{n}{M^2}\gamma^2\sum_{i>k^*}\lambda_i^2\right).
\end{aligned}$$

*b) Under Assumptions 3.2 - 3.4, it holds that*

$$Var\left(\bar{w}_{\frac{n}{M}}^{(m)}\right) := \frac{1}{2}\left\langle\mathbf{H}, \mathbb{E}\left[\bar{v}_n^{(m)} \otimes \bar{v}_n^{(m)}\right]\right\rangle_{op} \leq \frac{\sigma^2}{1 - \gamma\tau\,\mathrm{tr}(\mathbf{H})}\left(\frac{k^* M}{n} + \gamma^2\frac{n}{M} \cdot \sum_{i>k^*}\lambda_i^2\right).$$

**Lemma A.3.** *Set $k^* = \max\left\{k : \lambda_k \geq \frac{M}{n\gamma}\right\}$. If the step size satisfies $\gamma < 1/(\tau\,\mathrm{tr}(\mathbf{H}))$, we have for every $m \in [M]$:*

*a) Under Assumption 3.2 and 3.3, it holds that*

$$\begin{aligned}
Bias(\overline{\overline{w}}_M) \leq&\frac{M^2}{\gamma^2 n^2} \cdot \|\mathbf{w}_0 - \mathbf{w}^*\|_{\mathbf{H}_{0:k^*}^{-1}}^2 + \|\mathbf{w}_0 - \mathbf{w}^*\|_{\mathbf{H}_{k^*:\infty}^2} \\
&+ \frac{2\tau M^2\left(\|\mathbf{w}_0 - \mathbf{w}^*\|_{\mathbf{I}_{0:k^*}}^2 + \frac{n}{M}\gamma\|\mathbf{w}_0 - \mathbf{w}^*\|_{\mathbf{H}_{k^*:\infty}}^2\right)}{\gamma n(1 - \gamma\tau\,\mathrm{tr}(\mathbf{H}))} \cdot \left(\frac{k^*}{n} + \frac{n}{M^2}\gamma^2\sum_{i>k^*}\lambda_i^2\right).
\end{aligned}$$

*b) Under Assumptions 3.2 - 3.4 , it holds that*

$$Var(\overline{\overline{w}}_M) \leq \frac{\sigma^2}{1 - \gamma\tau\,\mathrm{tr}(\mathbf{H})}\left(\frac{k^*}{n} + \gamma^2\frac{n}{M^2}\cdot\sum_{i>k^*}\lambda_i^2\right).$$

*Proof of Lemma A.3 . a)* For the Bias-term we simply use

$$\mathrm{Bias}(\overline{\overline{w}}_M) = \frac{1}{2}\mathbb{E}\left\|\overline{\overline{b}}\right\|_{\mathbf{H}}^2 = \frac{1}{2}\mathbb{E}\left\|\frac{1}{M}\sum_{m=1}^{M}\bar{b}_{M\,n}^{(m)}\right\|_{\mathbf{H}}^2 \leq \frac{1}{M}\sum_{m=1}^{M}\frac{1}{2}\mathbb{E}\left\|\bar{b}_n^{(m)}\right\|_{\mathbf{H}}^2 = \frac{1}{M}\sum_{m=1}^{M}\mathrm{Bias}\left(\bar{w}_{\frac{n}{M}}^{(m)}\right). \quad (13)$$

Taking the bound of the local Bias-term $\mathrm{Bias}\left(\bar{w}_{\frac{n}{M}}^{(m)}\right)$ from A.2, proves the claim.

*b)* First we split the expectation operator as follows

$$\mathbb{E}\left[\overline{\overline{v}}_M \otimes \overline{\overline{v}}_M\right]$$

$$= \frac{1}{M^2}\sum_{m,m'=1}^{M}\mathbb{E}\left[\bar{v}_n^{(m)} \otimes \bar{v}_n^{(m')}\right]$$

$$= \frac{1}{M^2}\sum_{m=1}^{M}\mathbb{E}\left[\bar{v}_n^{(m)} \otimes \bar{v}_n^{(m)}\right] + \frac{1}{M^2}\sum_{m\neq m'}\mathbb{E}\left[\bar{v}_n^{(m)} \otimes \bar{v}_n^{(m')}\right]$$

$$=: I_1 + I_2. \quad (14)$$

Now we prove that the second operator $I_2$ is equal zero. First rewrite $I_2$ as

$$I_2 = \frac{1}{M^2}\sum_{m\neq m'}\frac{M^2}{n^2}\sum_{s,t=0}^{\frac{n}{M}-1}\mathbb{E}[v_t^{(m)} \otimes v_s^{(m')}].$$

Therefore it is enough to to prove $\mathbb{E}[v_t^{(m)} \otimes v_s^{(m')}] = 0$ for any $m \neq m'$. Since we assume our data sets to be independent we have $\mathbb{E}[v_t^{(m)} \otimes v_s^{(m')}] = \mathbb{E}[\langle\cdot, v_t^{(m)}\rangle]\mathbb{E}[v_s^{(m')}] = 0$, where the last equality follows from (7). This proves $I_2 = 0$. To sum up we have from (14) for the variance term,

$$\mathrm{Var}(\overline{\overline{w}}_M) = \frac{1}{2}\left\langle\mathbf{H}, \mathbb{E}\left[\overline{\overline{v}}_M \otimes \overline{\overline{v}}_M\right]\right\rangle_{op}$$

$$= \frac{1}{M^2}\sum_{m=1}^{M}\frac{1}{2}\left\langle\mathbf{H}, \mathbb{E}\left[\bar{v}_n^{(m)} \otimes \bar{v}_n^{(m)}\right]\right\rangle_{op}$$

$$= \frac{1}{M^2}\sum_{m=1}^{M}\mathrm{Var}\left(\bar{w}_{\frac{n}{M}}^{(m)}\right). \quad (15)$$

Using the bound of the local variance term from A.2 completes the proof. □

*Proof of Theorem 3.5.* Using lemma A.1 *a)* we have

$$\mathbb{E}\left[L(\overline{\overline{w}}_M)\right] - L(w^*) \leq 2\mathrm{Bias}(\overline{\overline{w}}_M) + 2\mathrm{Var}(\overline{\overline{w}}_M).$$

The claim now follows from lemma A.3. □

### A.2  Proofs Lower Bound

#### A.2.1  Lower Bound Bias

**Proposition A.4** (Lower Bound Bias). *Suppose Assumptions 3.2 and 3.6 are satisfied and let* $\gamma < \frac{1}{||\mathbf{H}||}$. *Recall the definition of* $\mathrm{Bias}(\overline{\overline{w}}_M)$ *in Lemma A.1. The bias of the distributed SGD estimator satisfies the lower bound*

$$\mathrm{Bias}(\overline{\overline{w}}_M) \geq \frac{M(M-1)}{100\gamma^2 n^2}\left(||w_1 - w^*||_{\mathbf{H}_{0:k^*}^{\dagger}}^2 + \frac{\gamma^2 n^2}{M^2}||w_1 - w^*||_{\mathbf{H}_{k^*:\infty}}^2\right).$$

*Proof of Proposition A.4.* From the definition of the bias in Lemma A.1, we have

$$\begin{aligned}
\mathrm{Bias}(\overline{\overline{w}}_M) &= \frac{1}{2}\left\langle \mathbf{H}, \mathbb{E}\left[\overline{\overline{b}}_M \otimes \overline{\overline{b}}_M\right]\right\rangle_{op} \\
&= \frac{1}{2M^2}\sum_{m_1=1}^{M}\sum_{m_2=1}^{M}\left\langle \mathbf{H}, \mathbb{E}\left[\overline{b}_n^{(m_1)} \otimes \overline{b}_n^{(m_2)}\right]\right\rangle_{op} \\
&= \frac{1}{2M^2}\sum_{m=1}^{M}\left\langle \mathbf{H}, \mathbb{E}\left[\overline{b}_n^{(m)} \otimes \overline{b}_n^{(m)}\right]\right\rangle_{op} + \frac{1}{2M^2}\sum_{m_1 \neq m_2}^{M}\left\langle \mathbf{H}, \mathbb{E}\left[\overline{b}_n^{(m_1)} \otimes \overline{b}_n^{(m_2)}\right]\right\rangle_{op}.
\end{aligned} \tag{16}$$

We show that the first term in the above decomposition can be lower bounded by zero. Indeed, from (C.2) and (C.4) in Zou et al. (2021b) we have for all $m = 1, ..., M$ the local lower bound

$$\begin{aligned}
\left\langle \mathbf{H}, \mathbb{E}\left[\overline{b}_n^{(m)} \otimes \overline{b}_n^{(m)}\right]\right\rangle_{op} &\geq \frac{M^2}{n^2}\sum_{t=1}^{\frac{n}{M}}\sum_{k=t}^{\frac{n}{M}}\left\langle (\mathbf{I} - \gamma\mathbf{H})^{k-t}\mathbf{H}, \mathbb{E}\left[b_t^{(m)} \otimes b_t^{(m)}\right]\right\rangle_{op} \\
&\geq \frac{M^2}{\gamma n^2}\left\langle \mathbf{I} - (\mathbf{I} - \gamma\mathbf{H})^{\frac{n}{2M}}, \mathbf{S}_{\frac{n}{2M}}^{(m)}\right\rangle_{op},
\end{aligned}$$

where we set

$$\mathbf{S}_{\frac{n}{2M}}^{(m)} := \sum_{t=1}^{\frac{n}{2M}}\mathbb{E}\left[b_t^{(m)} \otimes b_t^{(m)}\right].$$

Setting $\mathbf{B}_1 = b_1^{(m)} \otimes b_1^{(m)} = (w_1 - w^*) \otimes (w_1 - w^*)$ and applying Lemma C.4 from Zou et al. (2021b) gives then for all $m = 1, ..., M$

$$\mathbf{S}_{\frac{n}{2M}}^{(m)} \succeq \underbrace{\frac{\theta}{4}\mathrm{Tr}\left[(\mathbf{I} - (\mathbf{I} - \gamma\mathbf{H})^{\frac{n}{2M}})\mathbf{B}_1\right] \cdot ((\mathbf{I} - (\mathbf{I} - \gamma\mathbf{H})^{\frac{n}{2M}}))}_{PSD} + \underbrace{\sum_{t=1}^{\frac{n}{M}}(\mathbf{I} - \gamma\mathbf{H})^t \cdot \mathbf{B}_1 \cdot (\mathbf{I} - \gamma\mathbf{H})^t}_{PSD}$$

$$\succeq 0.$$

Hence,

$$\begin{aligned}
\frac{1}{2M^2}\sum_{m=1}^{M}\left\langle \mathbf{H}, \mathbb{E}\left[\overline{b}_n^{(m)} \otimes \overline{b}_n^{(m)}\right]\right\rangle_{op} &\geq \frac{1}{2M^2} \cdot \frac{M^2}{\gamma n^2}\sum_{m=1}^{M}\left\langle \mathbf{I} - (\mathbf{I} - \gamma\mathbf{H})^{\frac{n}{2M}}, \mathbf{S}_{\frac{n}{2M}}^{(m)}\right\rangle_{op} \\
&\geq 0.
\end{aligned} \tag{17}$$

We now bound the second term in equation 16. Note that by independence of the local nodes and with equation 6 we may write for any fixed $m_1 \neq m_2$

$$\mathbb{E}\left[\bar{b}_n^{(m_1)} \otimes \bar{b}_n^{(m_2)}\right] = \frac{M^2}{n^2} \sum_{t=1}^{\frac{n}{M}} \sum_{k=1}^{\frac{n}{M}} \mathbb{E}\left[b_t^{(m_1)}\right] \otimes \mathbb{E}\left[b_k^{(m_2)}\right]$$

$$= \frac{M^2}{n^2} \sum_{t=1}^{\frac{n}{M}} \sum_{k=1}^{\frac{n}{M}} (\mathbf{I} - \gamma\mathbf{H})^t \cdot \mathbf{B}_1 \cdot (\mathbf{I} - \gamma\mathbf{H})^k .$$

Hence,

$$\frac{1}{2M^2} \sum_{m_1 \neq m_2}^{M} \left\langle \mathbf{H}, \mathbb{E}\left[\bar{b}_n^{(m_1)} \otimes \bar{b}_n^{(m_2)}\right]\right\rangle_{op} = \frac{1}{2M^2} \frac{M^2}{n^2} \sum_{m_1 \neq m_2}^{M} \sum_{t=1}^{\frac{n}{M}} \sum_{k=1}^{\frac{n}{M}} \left\langle \mathbf{H}, (\mathbf{I} - \gamma\mathbf{H})^t \cdot \mathbf{B}_1 \cdot (\mathbf{I} - \gamma\mathbf{H})^k\right\rangle_{op}$$

$$= \frac{M(M-1)}{2\gamma n^2}\left\langle \sum_{k=1}^{\frac{n}{M}} (\mathbf{I} - \gamma\mathbf{H})^k \left(\mathbf{I} - (\mathbf{I} - \gamma\mathbf{H})^{\frac{n}{M}+1}\right), \mathbf{B}_1 \right\rangle_{op}$$

$$= \frac{M(M-1)}{2\gamma^2 n^2}\left\langle \left(\mathbf{I} - (\mathbf{I} - \gamma\mathbf{H})^{\frac{n}{M}+1}\right)^2 \mathbf{H}^{-1}, \mathbf{B}_1 \right\rangle_{op} .$$

Following now the lines of the proof of Lemma C.5 in Zou et al. (2021b) (adapted to our local setting) gives

$$\frac{1}{2M^2} \sum_{m_1 \neq m_2}^{M} \left\langle \mathbf{H}, \mathbb{E}\left[\bar{b}_n^{(m_1)} \otimes \bar{b}_n^{(m_2)}\right]\right\rangle_{op} \geq \frac{M(M-1)}{100\gamma^2 n^2}\left(||w_1 - w^*||_{\mathbf{H}_{0:k^*}^\dagger}^2 + \frac{\gamma^2 n^2}{M^2}||w_1 - w^*||_{\mathbf{H}_{k^*:\infty}}^2\right) .$$

Combining now the last bound with equation 17 and equation 16 finally gives

$$\text{Bias}(\bar{\bar{w}}_M) \geq \frac{M(M-1)}{100\gamma^2 n^2}\left(||w_1 - w^*||_{\mathbf{H}_{0:k^*}^\dagger}^2 + \frac{\gamma^2 n^2}{M^2}||w_1 - w^*||_{\mathbf{H}_{k^*:\infty}}^2\right) .$$

$$\square$$

### A.2.2 Lower Bound Variance

**Proposition A.5** (Lower Bound Variance). *Suppose Assumptions 3.2 and 3.6 are satisfied and let $\frac{n}{M} \geq 500$, $\gamma < \frac{1}{||\mathbf{H}||}$. Recall the definition of $\text{Var}(\bar{\bar{w}}_M)$ in Lemma A.1. The variance of the distributed SGD estimator satisfies the lower bound*

$$\text{Var}(\bar{\bar{w}}_M) \geq \frac{\sigma_{noise}^2}{100} \cdot \left(\frac{k^*}{n} + \frac{\gamma^2 n}{M^2}\sum_{j>k^*} \lambda_j^2\right) .$$

*Proof of Proposition A.5.* From the definition of the variance in Lemma A.1, we have

$$\text{Var}(\bar{\bar{w}}_M) = \frac{1}{2}\left\langle \mathbf{H}, \mathbb{E}\left[\bar{\bar{v}}_M \otimes \bar{\bar{v}}_M\right]\right\rangle_{op}$$

$$= \frac{1}{2M^2} \sum_{m_1=1}^{M} \sum_{m_2=1}^{M} \left\langle \mathbf{H}, \mathbb{E}\left[\bar{v}_n^{(m_1)} \otimes \bar{v}_n^{(m_2)}\right]\right\rangle_{op}$$

$$= \frac{1}{2M^2} \sum_{m=1}^{M} \left\langle \mathbf{H}, \mathbb{E}\left[\bar{v}_n^{(m)} \otimes \bar{v}_n^{(m)}\right]\right\rangle_{op} + \frac{1}{2M^2} \sum_{m_1 \neq m_2}^{M} \left\langle \mathbf{H}, \mathbb{E}\left[\bar{v}_n^{(m_1)} \otimes \bar{v}_n^{(m_2)}\right]\right\rangle_{op} . \qquad (18)$$

We first lower bound the first term. By Eq. (C.3) and Lemma C.3 in Zou et al. (2021b) (adapted to our local setting) we obtain

$$\frac{1}{2M^2} \sum_{m=1}^{M} \left\langle \mathbf{H}, \mathbb{E}\left[ \bar{v}_n^{(m)} \otimes \bar{v}_n^{(m)} \right] \right\rangle_{op} \geq \frac{1}{2M^2} \frac{M^2}{n^2} \sum_{m=1}^{M} \sum_{t=0}^{\frac{n}{M}-1} \sum_{k=t}^{\frac{n}{M}-1} \left\langle (\mathbf{I} - \gamma\mathbf{H})^{k-t}\mathbf{H}, \mathbb{E}\left[ v_t^{(m)} \otimes v_t^{(m)} \right] \right\rangle_{op}$$

$$\geq \frac{\sigma_{noise}^2}{100M^2} \sum_{m=1}^{M} \left( \frac{M}{n} k^* + \frac{\gamma^2 n}{M} \sum_{j>k^*} \lambda_j^2 \right)$$

$$= \frac{\sigma_{noise}^2}{100} V_{k^*}(n, M) \, ,$$

where

$$V_{k^*}(n, M) := \left( \frac{k^*}{n} + \frac{\gamma^2 n}{M^2} \sum_{j>k^*} \lambda_j^2 \right) \, .$$

To derive the final bound we argue that the second term in equation 18 is zero. Indeed, by independence of the local nodes we may write for any $m_1 \neq m_2$ with equation 7

$$\mathbb{E}\left[ v_t^{(m_1)} \otimes v_k^{(m_2)} \right] = \mathbb{E}\left[ v_t^{(m_1)} \right] \otimes \mathbb{E}\left[ v_k^{(m_2)} \right]$$

$$= (\mathbf{I} - \gamma\mathbf{H})^t (v_0^{(m_1)} \otimes v_0^{(m_2)})(\mathbf{I} - \gamma\mathbf{H})^k$$

$$= 0 \, ,$$

since $v_0^{(m)} = 0$ for all $m = 1, ..., M$. Hence,

$$\frac{1}{2M^2} \sum_{m_1 \neq m_2}^{M} \left\langle \mathbf{H}, \mathbb{E}\left[ \bar{v}_n^{(m_1)} \otimes \bar{v}_n^{(m_2)} \right] \right\rangle_{op} = 0 \, .$$

this finishes the proof. $\qquad\square$

### A.3   Proofs Rates of Convergence

*Proof of Corollary 3.9.* Let the sequence $M_n \leq \sqrt{\frac{\gamma(1-2\gamma\tau)n}{R^2}}$. By definition of $k^*$ we know that $k^* = \tilde{d} = \left( \frac{n}{M_n} \right)^r$ and hence $\lambda_{k^*} = \left( \frac{M_n}{n} \right)^r$. We first bound the bias from Theorem 3.5. Since $||w^*||_2 \leq R$ by assumption, we find

$$||w^*||_{\mathbf{H}_{0:k^*}^\dagger}^2 \leq \frac{||w^*||_2^2}{\lambda_{k^*}} \leq R^2 \left( \frac{n}{M_n} \right)^r \, . \tag{19}$$

Similarly, since $\frac{n}{M_n} \to \infty$ as $n \to \infty$, there exists $n_0 \in \mathbb{N}$ such that

$$||w^*||_{\mathbf{H}_{k^*:\infty}}^2 \leq \frac{R^2}{\left( \frac{n}{M_n} \right)^q - \left( \frac{n}{M_n} \right)^r} \leq c_{n_0} R^2 \left( \frac{M_n}{n} \right)^q \, , \tag{20}$$

for any $n \geq n_0$ and some $c_{n_0} < \infty$. Using that $\text{Tr}[\mathbf{H}] = 2$ and $||w^*||_{\mathbf{I}_{0:k^*}}^2 \leq R^2$, we find for all $n \geq n_0$, for some $n_0 \in \mathbb{N}$, that

$$\frac{2\tau M^2 \left( ||w^*|_{\mathbf{I}_{0:k}}^2 + \gamma \frac{n}{M} ||w^*||_{\mathbf{H}_{k:\infty}}^2 \right)}{\gamma n (1 - \gamma\tau \text{Tr}[\mathbf{H}])} \leq 4 \max\{1, c_{n_0}\} \frac{\tau R^2}{1 - 2\gamma\tau} \frac{M_n^2}{\gamma n} \, .$$

Note that we also use that $M_n \leq n$ and hence $\left( \frac{M_n}{n} \right)^{q-1} \leq 1$, since $q > 1$. Since

$$M_n \leq \sqrt{\frac{\gamma(1 - 2\gamma\tau)n}{R^2}}$$

we have

$$\frac{\tau R^2}{1 - 2\gamma\tau} \frac{M_n^2}{\gamma n} \leq 1$$

and hence

$$\frac{2\tau M_n^2 \left( ||w^*|_{\mathbf{I}_{0:k}}^2 + \gamma \frac{n}{M} ||w^*||_{\mathbf{H}_{k:\infty}}^2 \right)}{\gamma n (1 - \gamma\tau \operatorname{Tr}[\mathbf{H}])} \leq 4 \max\{1, c_{n_0}\} \, . \tag{21}$$

We further observe that by the definition of the spectrum of $\mathbf{H}$

$$\sum_{j>k^*} \lambda_l^2 = \sum_{j=\tilde{d}}^{d} \frac{1}{d - \tilde{d}} = \frac{1}{\left(\frac{n}{M_n}\right)^q - \left(\frac{n}{M_n}\right)^r} \leq c_{n_0} \left(\frac{M_n}{n}\right)^q ,$$

for any $n$ sufficiently large, by using the argumentation as above. Hence,

$$V_{k^*}(n, M_n) := \frac{k^*}{n} + \gamma^2 \frac{n}{M_n^2} \sum_{j=k^*+1}^{\infty} \lambda_j^2$$

$$\leq \max\{1, c_{n_0}\} \frac{1}{M_n} \cdot \left( \left(\frac{M_n}{n}\right)^{1-r} + \gamma^2 \left(\frac{M_n}{n}\right)^{q-1} \right) . \tag{22}$$

Combining (19), (20), (21) and (22), we find for the bias term

$$\operatorname{Bias}(M_n) \leq \frac{R^2}{\gamma^2 (n/M_n)^2} \left(\frac{n}{M_n}\right)^r + c_{n_0} R^2 \left(\frac{M_n}{n}\right)^q + 4 \max\{1, c_{n_0}\} V_{k^*}(n, M_n)$$

$$\leq \max\{1, c_{n_0}\} \, R^2 \left( \frac{1}{\gamma^2} \left(\frac{M_n}{n}\right)^{2-r} + \left(\frac{M_n}{n}\right)^q \right) + \tag{23}$$

$$4 \max\{1, c_{n_0}\}^2 \frac{1}{M_n} \cdot \left( \left(\frac{M_n}{n}\right)^{1-r} + \gamma^2 \left(\frac{M_n}{n}\right)^{q-1} \right) . \tag{24}$$

We now turn to the bound of the variance term. From equation 22 we have

$$\operatorname{Var}(M_n) \leq \max\{1, c_{n_0}\} \left( \frac{\sigma^2}{1 - \gamma\tau \operatorname{Tr}[\mathbf{H}]} \right) \cdot \frac{1}{M_n} \cdot \left( \left(\frac{M_n}{n}\right)^{1-r} + \gamma^2 \left(\frac{M_n}{n}\right)^{q-1} \right) .$$

Combining the bounds for bias and variance leads to the total error bound

$$\mathbb{E}\left[L(\overline{\overline{w}}_{M_n})\right] - L(w^*) \leq$$

$$2\tilde{c}_{n_0} \cdot c_{\gamma,\tau,\sigma} \cdot \left( R^2 \left( \frac{1}{\gamma^2} \left(\frac{M_n}{n}\right)^{2-r} + \left(\frac{M_n}{n}\right)^q \right) + \cdot \frac{1}{M_n} \cdot \left( \left(\frac{M_n}{n}\right)^{1-r} + \gamma^2 \left(\frac{M_n}{n}\right)^{q-1} \right) \right) ,$$

with

$$c_{\gamma,\tau,\sigma} := 1 + \frac{\sigma^2}{1 - \gamma\tau \operatorname{Tr}[\mathbf{H}]} , \quad \tilde{c}_{n_0} = 4 \max\{1, c_{n_0}\}^2 ,$$

holding for any $n$ sufficiently large. We proceed by further simplifying the right hand side of the above inequality. Since $\tau \geq 1$ and $1 - \gamma\tau \operatorname{Tr}[\mathbf{H}] < 1$, the assumption on $M_n$ implies that

$$M_n^2 \leq \frac{n\gamma}{R^2} ,$$

further implying that

$$\frac{R^2}{\gamma} \left(\frac{M_n}{n}\right)^{2-r} \leq \frac{1}{M_n} \left(\frac{M_n}{n}\right)^{1-r}$$

and

$$R^2 \left( \frac{M_n}{n} \right)^q \leq \frac{\gamma}{M_n} \left( \frac{M_n}{n} \right)^{q-1} .$$

As a result, applying Theorem 3.5, the excess risk can be bounded by

$$\mathbb{E}\left[L(\overline{\overline{w}}_{M_n})\right] - L(w^*) \leq 4\tilde{c}_{n_0} \cdot c_{\gamma,\tau,\sigma} \cdot \frac{1}{M_n} \left( \left( \frac{1}{\gamma} + 1 \right) \left( \frac{M_n}{n} \right)^{1-r} + (\gamma + \gamma^2) \left( \frac{M_n}{n} \right)^{q-1} \right)$$

$$\leq 4\tilde{c}_{n_0} \cdot c_{\gamma,\tau,\sigma} \cdot \frac{1}{\gamma M_n} \left( \left( \frac{M_n}{n} \right)^{1-r} + \left( \frac{M_n}{n} \right)^{q-1} \right) .$$

In the last step we use that $\gamma < \frac{1}{2\tau} < \frac{1}{2}$. $\qquad\square$

*Proof of Corollary 3.10.* Assume the sequence $(M_n)_n$ satisfies $M_n/n \to 0$ as $n \to \infty$. We use Theorem 3.5 to bound the excess risk and find estimates for bias and variance. By the definition of $k^*$ we have

$$k^* = \max\left\{ k \in \mathbb{N} : k \leq \left( \frac{\gamma n}{M_n} \right)^{\frac{1}{1+r}} \right\} = \left\lfloor \left( \frac{\gamma n}{M_n} \right)^{\frac{1}{1+r}} \right\rfloor .$$

Hence, there exists $n_0 \in \mathbb{N}$ such that for all $n \geq n_0$

$$c_{n_0} \left( \frac{\gamma n}{M_n} \right)^{\frac{1}{1+r}} \leq k^* \leq C_{n_0} \left( \frac{\gamma n}{M_n} \right)^{\frac{1}{1+r}} ,$$

for some constants $0 < c_{n_0} \leq C_{n_0}$. Therefore,

$$\lambda_{k^*} = (k^*)^{-(1+r)} \leq \left( \frac{1}{c_{n_0}} \right)^{1+r} \cdot \frac{M_n}{\gamma n}$$

and

$$\frac{1}{\lambda_{k^*}} = (k^*)^{1+r} \leq C_{n_0}^{1+r} \cdot \frac{n}{\gamma M_n} .$$

We therefore get for the first two terms of the bias

$$\frac{M_n^2}{\gamma^2 n^2} \cdot \|w^*\|_{\mathbf{H}_{0:k^*}^\dagger}^2 \leq \frac{R^2 M_n^2}{\gamma^2 n^2 \lambda_{k^*}} \tag{25}$$

$$\leq C_{n_0}^{1+r} \frac{R^2}{\gamma} \cdot \frac{M_n}{n} . \tag{26}$$

and

$$\|w^*\|_{\mathbf{H}_{k^*:\infty}}^2 \leq R^2 \lambda_{k^*} \leq R^2 \left( \frac{1}{c_{n_0}} \right)^{1+r} \cdot \frac{M_n}{\gamma n} . \tag{27}$$

We now bound the last term of the bias. To this end, we apply a well known bound for sums over decreasing functions, i.e.,

$$\sum_{j \geq k} f(j) \leq \int_k^\infty f(x) dx .$$

This gives

$$\sum_{j > k^*} \lambda_j^2 \leq \int_{k^*}^\infty x^{-2(r+1)} dx \leq \frac{1}{2r+1} (k^*)^{-(2r+1)} \leq \frac{1}{2r+1} c_{n_0}^{-(2r+1)} \left( \frac{M_n}{\gamma n} \right)^{1+\frac{r}{1+r}} .$$

Thus,

$$
\begin{aligned}
V_{k^*}(n, M_n) &= \frac{k^*}{n} + \gamma^2 \frac{n}{M^2} \sum_{j=k^*+1}^{\infty} \lambda_j^2 \\
&\leq \frac{1}{n} C_{n_0} \left( \frac{\gamma n}{M_n} \right)^{\frac{1}{1+r}} + \gamma^2 \frac{n}{M_n^2} \frac{c_{n_0}^{-(2r+1)}}{2r+1} \left( \frac{M_n}{\gamma n} \right)^{1 + \frac{r}{1+r}} \\
&\leq c'_{r,n_0} \left( \frac{1}{n} \left( \frac{\gamma n}{M_n} \right)^{\frac{1}{1+r}} + \frac{\gamma}{M_n} \left( \frac{M_n}{\gamma n} \right)^{\frac{r}{1+r}} \right) \\
&\leq 2 c'_{r,n_0} \cdot \frac{\gamma}{M_n} \left( \frac{M_n}{\gamma n} \right)^{\frac{r}{1+r}} ,
\end{aligned}
\tag{28}
$$

with

$$
c'_{r,n_0} = \max \left\{ C_{n_0}, \frac{c_{n_0}^{-(2r+1)}}{2r+1} \right\} .
$$

Moreover,

$$
\begin{aligned}
\frac{2\tau M_n^2 \left( \|w^*\|_{\mathbf{I}_{0:k^*}}^2 + \gamma \frac{n}{M_n} \|w^*\|_{\mathbf{H}_{k^*:\infty}}^2 \right)}{\gamma n (1 - \gamma\tau \operatorname{Tr}[\mathbf{H}])} &\leq \frac{2\tau M_n^2}{\gamma n (1 - \gamma\tau \operatorname{Tr}[\mathbf{H}])} \left( R^2 + R^2 \gamma \frac{n}{M_n} \lambda_{k^*} \right) \\
&\leq c''_{n_0} \frac{2\tau M_n^2}{\gamma n (1 - \gamma\tau \operatorname{Tr}[\mathbf{H}])} R^2 \left( 1 + \gamma \frac{n}{M_n} \frac{M_n}{\gamma n} \right) \\
&\leq 2 c''_{n_0} \frac{2\tau}{(1 - \gamma\tau \operatorname{Tr}[\mathbf{H}])} \cdot \frac{R^2 M_n^2}{\gamma n} ,
\end{aligned}
$$

with

$$
c''_{n_0} = \max \left\{ 1, \left( \frac{1}{c_{n_0}} \right)^{1+r} \right\} .
$$

Hence, combining this with equation 28 and choosing

$$
M_n \leq \frac{\sqrt{\gamma n}}{R}
\tag{29}
$$

leads to

$$
\begin{aligned}
& \frac{2\tau M_n^2 \left( \|w^*\|_{\mathbf{I}_{0:k^*}}^2 + \gamma \frac{n}{M_n} \|w^*\|_{\mathbf{H}_{k^*:\infty}}^2 \right)}{\gamma n (1 - \gamma\tau \operatorname{Tr}[\mathbf{H}])} \cdot V_{k^*}(n, M_n) \\
&\leq 2 c''_{n_0} \frac{2\tau}{(1 - \gamma\tau \operatorname{Tr}[\mathbf{H}])} \cdot \frac{R^2 M_n^2}{\gamma n} \cdot 2 c'_{r,n_0} \cdot \frac{\gamma}{M_n} \left( \frac{M_n}{\gamma n} \right)^{\frac{r}{1+r}} \\
&\leq c_{r,n_0} \frac{\tau}{(1 - \gamma\tau \operatorname{Tr}[\mathbf{H}])} \cdot \frac{R^2 M_n^2}{\gamma n} \cdot \frac{\gamma}{M_n} \left( \frac{M_n}{\gamma n} \right)^{\frac{r}{1+r}} \\
&\leq c_{r,n_0} \frac{\tau}{(1 - \gamma\tau \operatorname{Tr}[\mathbf{H}])} \cdot \frac{\gamma}{M_n} \left( \frac{M_n}{\gamma n} \right)^{\frac{r}{1+r}} ,
\end{aligned}
\tag{30}
$$

where $c_{r,n_0} = 8 c''_{n_0} \cdot c'_{r,n_0}$. Combining (26), (27) and (30), we find for all $n \geq n_0$

$$
\operatorname{Bias}(M_n) \leq \tilde{c}_{r,n_0} \cdot \frac{R^2}{\gamma} \cdot \frac{M_n}{n} + c_{r,n_0} \frac{\tau}{(1 - \gamma\tau \operatorname{Tr}[\mathbf{H}])} \cdot \frac{\gamma}{M_n} \left( \frac{M_n}{\gamma n} \right)^{\frac{r}{1+r}} ,
\tag{31}
$$

where we set

$$
\tilde{c}_{r,n_0} = 2 \max \left\{ C_{n_0}^{1+r}, \left( \frac{1}{c_{n_0}} \right)^{1+r} \right\} .
$$

We now turn to bounding the variance. Using equation 28 once more, the variance can be bounded by

$$\mathrm{Var}(M_n) \leq \frac{\sigma^2}{1 - \gamma\tau\,\mathrm{Tr}[\mathbf{H}]} \cdot 2c'_{r,n_0} \cdot \frac{\gamma}{M_n}\left(\frac{M_n}{\gamma n}\right)^{\frac{r}{1+r}} .$$

Combining the bias bound equation 31 with the variance bound, we obtain for the excess risk

$$\mathbb{E}\left[L(\overline{\overline{w}}_{M_n})\right] - L(w^*) \leq c_{r,n_0,\tau,\sigma}\left(\frac{R^2}{\gamma} \cdot \frac{M_n}{n} + \frac{\gamma}{M_n}\left(\frac{M_n}{\gamma n}\right)^{\frac{r}{1+r}}\right) .$$

$$c_{r,n_0,\tau,\sigma} := \max\left\{\tilde{c}_{r,n_0}, 2c_{r,n_0} \cdot c'_{r,n_0} \cdot \frac{\max\{\tau,\sigma^2\}}{1 - \gamma\tau\,\mathrm{Tr}[\mathbf{H}]}\right\} .$$

Note that the choice

$$M_n \leq \left(\frac{\gamma}{R^2}\right)^{\frac{1+r}{2+r}} \cdot (\gamma n)^{\frac{1}{2+r}} \tag{32}$$

leads to a dominating variance part, i.e.

$$\frac{R^2}{\gamma} \cdot \frac{M_n}{n} \leq \frac{\gamma}{M_n}\left(\frac{M_n}{\gamma n}\right)^{\frac{r}{1+r}}$$

and

$$\mathbb{E}\left[L(\overline{\overline{w}}_{M_n})\right] - L(w^*) \leq 2c_{r,n_0,\tau,\sigma}\frac{\gamma}{M_n}\left(\frac{M_n}{\gamma n}\right)^{\frac{r}{1+r}} .$$

Note that the choice equation 32 is compatible with the choice equation 29, i.e.,

$$M_n \leq \left(\frac{\gamma}{R^2}\right)^{\frac{1+r}{2+r}} \cdot (\gamma n)^{\frac{1}{2+r}} \leq \frac{\sqrt{\gamma n}}{R} ,$$

following from the fact that $r > 0$, provided that $n$ is sufficiently large. □

# B  PROOFS SECTION 4 ( COMPARISON OF SAMPLE COMPLEXITY OF DSGD AND DRR)

## B.1  Lower Bound for distributed ridge regression

In this section we derive a lower bound for the distributed RR estimator. We adopt the following notation and assumptions from Tsigler & Bartlett (2020).

- $\mathbf{H}^{-1/2}x$, where $x \in \mathbb{R}^d$ is sub-Gaussian with independent components

- $\mathbf{X} = (\sqrt{\lambda_1}z_1, ..., \sqrt{\lambda_d}z_d)$ with $z_j$ being sub-Gaussian with independent components

- $\mathbf{A} := \mathbf{X}\mathbf{X}^T + \lambda I_n, \quad \mathbf{A}_m := \mathbf{X}_m\mathbf{X}_m^T + \lambda I_n$

- $\mathbf{A}_{-j} = \sum_{i \neq j}\lambda_i z_i z_i^T + \lambda I_n$

Crucial for our analysis is the following quantity, called the *local effective dimension* for the RR problem:

$$k^*_{\mathrm{RR}} := \min\left\{k : \lambda_{k+1} \leq \frac{M\left(\lambda + \sum_{j>k}\lambda_j\right)}{bn}\right\} . \tag{33}$$

### B.1.1 Bias-Variance Decomposition DRR

**Definition B.1** (Bias and Variance of Distributed RR). *Let*

$$\Pi_m(\lambda) := \left(\mathbf{X}_m^T \mathbf{X}_m + \lambda\right)^{-1} \mathbf{X}_m^T \mathbf{X}_m - Id \,,$$

$$\widehat{\text{Bias}}(\overline{w}_n^{\text{RR}}(\lambda)) := \left\| \mathbf{H}^{1/2} \left( \frac{1}{M} \sum_{m=1}^M \Pi_m(\lambda) w^* \right) \right\|^2 \,,$$

$$\widehat{\text{Var}}(\overline{w}_n^{\text{RR}}(\lambda)) := \left\| \mathbf{H}^{1/2} \left( \frac{1}{M} \sum_{m=1}^M (\mathbf{X}_m^T \mathbf{X}_m + \lambda)^{-1} \mathbf{X}_m^T \epsilon_m \right) \right\|^2 \,.$$

*We call*

$$\text{Bias}(\overline{w}_n^{\text{RR}}(\lambda)) = \mathbb{E}\left[ \widehat{\text{Bias}}(\overline{w}_n^{\text{RR}}(\lambda)) \right]$$

*the (expected) bias of the distributed RR estimator and*

$$\text{Var}(\overline{w}_n^{\text{RR}}(\lambda)) = \mathbb{E}\left[ \widehat{\text{Var}}(\overline{w}_n^{\text{RR}}(\lambda)) \right]$$

*the (expected) variance.*

We immediately obtain:

**Lemma B.2.** *The excess risk satisfies*

$$\mathbb{E}\left[ ||\mathbf{H}^{1/2}(\overline{w}_n^{\text{RR}}(\lambda) - w^*)||^2 \right] = \text{Bias}(\overline{w}_n^{\text{RR}}(\lambda)) + \text{Var}(\overline{w}_n^{\text{RR}}(\lambda)) \,.$$

*Proof of Lemma B.2.* We split the excess risk as

$$\begin{aligned}
||\mathbf{H}^{1/2}(\overline{w}_n^{\text{RR}}(\lambda) - w^*)||^2 &= \left\| \mathbf{H}^{1/2} \left( \frac{1}{M} \sum_{m=1}^M \hat{w}_m^{\text{RR}}(\lambda) - w^* \right) \right\|^2 \\
&= \left\| \mathbf{H}^{1/2} \left( \frac{1}{M} \sum_{m=1}^M (\mathbf{X}_m^T \mathbf{X}_m + \lambda)^{-1} \mathbf{X}_m^T \mathbf{Y}_m - w^* \right) \right\|^2 \\
&= \left\| \mathbf{H}^{1/2} \left( \frac{1}{M} \sum_{m=1}^M (\mathbf{X}_m^T \mathbf{X}_m + \lambda)^{-1} \mathbf{X}_m^T (\mathbf{X}_m w^* + \epsilon_m) - w^* \right) \right\|^2 \\
&= \widehat{\text{Bias}}(\overline{w}_n^{\text{RR}}(\lambda)) + \widehat{\text{Var}}(\overline{w}_n^{\text{RR}}(\lambda)) \\
&\quad + \frac{2}{M^2} \sum_{m=1}^M \sum_{m'=1}^M \left\langle \mathbf{H} \, \Pi_m(\lambda) w^*, (\mathbf{X}_m^T \mathbf{X}_m + \lambda)^{-1} \mathbf{X}_m^T \epsilon_m \right\rangle \,.
\end{aligned}$$

We argue that the expectation with respect to the noise (i.e. conditioned on $\mathbf{X}$) of the last term is equal to zero. Indeed, by linearity and since $\epsilon_m$ is centered (conditioned on $\mathbf{X}_m$) for all $m \in [M]$, we find

$$\begin{aligned}
\mathbb{E}_{\epsilon_m} \left[ \left\langle \mathbf{H} \, \Pi_m(\lambda) w^*, (\mathbf{X}_m^T \mathbf{X}_m + \lambda)^{-1} \mathbf{X}_m^T \epsilon_m \right\rangle \right] &= \left\langle \mathbf{H} \, \Pi_m(\lambda) w^*, (\mathbf{X}_m^T \mathbf{X}_m + \lambda)^{-1} \mathbf{X}_m^T \mathbb{E}_{\epsilon_m}[\epsilon_m] \right\rangle \\
&= 0 \,.
\end{aligned}$$

Hence,

$$\mathbb{E}\left[ ||\mathbf{H}^{1/2}(\overline{w}_n^{\text{RR}}(\lambda) - w^*)||^2 \right] = \mathbb{E}\left[ \widehat{\text{Bias}}(\overline{w}_n^{\text{RR}}(\lambda)) \right] + \mathbb{E}\left[ \widehat{\text{Var}}(\overline{w}_n^{\text{RR}}(\lambda)) \right] \,.$$

$\square$

### B.1.2 Lower Bound of Bias for DRR

**Proposition B.3** (Lower Bound of Bias for local RR). *Assume $\mathbf{H}$ is strictly positive definite with $\mathrm{Tr}[H] < \infty$. There exist absolute constants $b > 1$, $c > 1$ such that*

$$\mathrm{Bias}(\overline{w}_n^{\mathrm{RR}}(\lambda)) \geq \frac{M-1}{cM} \cdot \left( \frac{M^2\left(\lambda + \sum_{j > k_{\mathrm{RR}}^*} \lambda_j\right)^2}{n^2} \cdot ||w^*||_{\mathbf{H}_{0:k_{\mathrm{RR}}^*}^{-1}}^2 + ||w^*||_{\mathbf{H}_{k_{\mathrm{RR}}^*:\infty}}^2 \right) ,$$

*where $k_{\mathrm{RR}}^*$ is defined in equation 33.*

For proving this Proposition we need the following Lemma.

**Lemma B.4.** *Let $\tilde{\mathbf{X}} \in \mathbb{R}^{n \times d}$ be an independent copy of $\mathbf{X} \in \mathbb{R}^{n \times d}$ and set $\tilde{\mathbf{A}} = \tilde{\mathbf{X}}\tilde{\mathbf{X}}^T + \lambda$. Define further the operator*

$$\mathbf{B} := (I_d - \mathbf{X}^T \mathbf{A}^{-1} \mathbf{X})\mathbf{H}(I_d - \tilde{\mathbf{X}}^T \tilde{\mathbf{A}}^{-1} \tilde{\mathbf{X}}) .$$

*1. For any $i \neq j$, we have*

$$\mathbb{E}_{\mathbf{X},\tilde{\mathbf{X}}}[\mathbf{B}_{ij}] = 0 .$$

*2. The diagonal elements satisfy for any $k$*

$$\mathbb{E}_{\mathbf{X},\tilde{\mathbf{X}}}[\mathbf{B}_{ii}] \geq \frac{1}{c} \cdot \frac{\lambda_i}{\left(1 + \frac{\lambda_i}{\lambda_{k+1}} \cdot \frac{n}{\rho_k}\right)^2} ,$$

*for some absolute constant $c > 1$ and where we define*

$$\rho_k = \frac{\lambda + \sum_{j > k} \lambda_j}{\lambda_{k+1}} .$$

*Proof of Lemma B.4.* Recall that $\mathbf{H} = diag\{\lambda_1, ..., \lambda_d\}_j$ and

$$\mathbf{X} = (\sqrt{\lambda_1}z_1, ..., \sqrt{\lambda_d}z_d) , \quad \tilde{\mathbf{X}} = (\sqrt{\lambda_1}\tilde{z}_1, ..., \sqrt{\lambda_d}\tilde{z}_d) .$$

1. Let $i \neq j$. We expand

$$\mathbf{B}_{ij} = \underbrace{\langle e_i, \mathbf{H}e_j \rangle}_{=0} - \sqrt{\lambda_i}\langle e_j, \mathbf{H}\mathbf{X}^T \mathbf{A}^{-1} z_i \rangle - \sqrt{\lambda_j}\langle e_i, \mathbf{H}\tilde{\mathbf{X}}^T \tilde{\mathbf{A}}^{-1} \tilde{z}_j \rangle + \sqrt{z_i z_j}\langle z_i, \mathbf{A}^{-1}\mathbf{X}\mathbf{H}\tilde{\mathbf{X}}^T \tilde{\mathbf{A}}^{-1} \tilde{z}_j \rangle$$

$$= -\lambda_j \sqrt{\lambda_i \lambda_j}\langle z_j, \mathbf{A}^{-1} z_i \rangle - \lambda_i \sqrt{\lambda_i \lambda_j}\langle \tilde{z}_i, \tilde{\mathbf{A}}^{-1} \tilde{z}_j \rangle + \sqrt{\lambda_i \lambda_j}\langle z_i, \mathbf{A}^{-1}\mathbf{X}\mathbf{H}\tilde{\mathbf{X}}^T \tilde{\mathbf{A}}^{-1} \tilde{z}_j \rangle .$$

We define the map $F(z_j) := \langle z_j, \mathbf{A}^{-1} z_i \rangle$. Following the lines of the proof of Lemma C.7 in Zou et al. (2021a) shows that $\mathbb{E}_{z_j}[F(z_j)] = 0$. Using similar arguments, the same is true for the second and last term, showing the result.

2. We expand

$$\mathbf{B}_{ii} = \left\langle \mathbf{H}(e_i - \sqrt{\lambda_i}\mathbf{X}^T \mathbf{A}^{-1} z_i), e_i - \sqrt{\lambda_i}\tilde{\mathbf{X}}^T \tilde{\mathbf{A}}^{-1} \tilde{z}_i \right\rangle$$

$$= \underbrace{\langle \mathbf{H}e_i, e_i \rangle}_{=\lambda_i} + \lambda_i \langle \mathbf{H}\mathbf{X}^T \mathbf{A}^{-1} z_i, \tilde{\mathbf{X}}^T \tilde{\mathbf{A}}^{-1} \tilde{z}_i \rangle - \sqrt{\lambda_i}\langle \mathbf{H}e_i, \tilde{\mathbf{X}}^T \tilde{\mathbf{A}}^{-1} \tilde{z}_i \rangle - \sqrt{\lambda_i}\langle \mathbf{H}e_i, \mathbf{X}^T \mathbf{A}^{-1} z_i \rangle$$

$$= \lambda_i \left[1 - \lambda_i\left(\langle z_i, \mathbf{A}^{-1} z_i \rangle + \langle \tilde{z}_i, \tilde{\mathbf{A}}^{-1} \tilde{z}_i \rangle\right)\right] + \lambda_i \langle \mathbf{H}\mathbf{X}^T \mathbf{A}^{-1} z_i, \tilde{\mathbf{X}}^T \tilde{\mathbf{A}}^{-1} \tilde{z}_i \rangle .$$

Setting

$$a_i := \langle z_i, \mathbf{A}^{-1} z_i \rangle , \quad \tilde{a}_i := \langle \tilde{z}_i, \tilde{\mathbf{A}}^{-1} \tilde{z}_i \rangle$$

we further find that

$$\lambda_i \langle \mathbf{H}\mathbf{X}^T \mathbf{A}^{-1} z_i, \tilde{\mathbf{X}}^T \tilde{\mathbf{A}}^{-1} \tilde{z}_i \rangle = \lambda_i \sum_{j=1}^{d} \lambda_j (\mathbf{X}^T \mathbf{A}^{-1} z_i)_j \cdot (\tilde{\mathbf{X}}^T \tilde{\mathbf{A}}^{-1} \tilde{z}_i)_j$$

$$= \lambda_i \sum_{j=1}^{d} \lambda_j^2 \langle z_j, \mathbf{A}^{-1} z_i \rangle \cdot \langle \tilde{z}_j, \tilde{\mathbf{A}}^{-1} z_i \rangle$$

$$= \lambda_i^3 \cdot a_i \cdot \tilde{a}_i + \lambda_i \sum_{j \neq i} \lambda_j^2 \langle z_j, \mathbf{A}^{-1} z_i \rangle \cdot \langle \tilde{z}_j, \tilde{\mathbf{A}}^{-1} z_i \rangle .$$

By independence, the last term is non-negative in expectation, i.e.

$$\mathbb{E}_{\mathbf{X},\tilde{\mathbf{X}}} \left[ \lambda_i \sum_{j \neq i} \lambda_j^2 \langle z_j, \mathbf{A}^{-1} z_i \rangle \cdot \langle \tilde{z}_j, \tilde{\mathbf{A}}^{-1} z_i \rangle \right]$$

$$= \lambda_i \sum_{j \neq i} \lambda_j^2 \mathbb{E}_{\mathbf{X}} \left[ \langle z_j, \mathbf{A}^{-1} z_i \rangle \right] \cdot \mathbb{E}_{\tilde{\mathbf{X}}} \left[ \langle \tilde{z}_j, \tilde{\mathbf{A}}^{-1} z_i \rangle \right]$$

$$= \lambda_i \sum_{j \neq i} \lambda_j^2 \cdot \mathbb{E}_{\mathbf{X}} \left[ \langle z_j, \mathbf{A}^{-1} z_i \rangle \right]^2$$

$$\geq 0 .$$

Hence, for deriving a lower bound in expectation it is sufficient to lower bound the expression

$$\lambda_i \cdot [1 - \lambda_i(a_i + \tilde{a}_i)] + \lambda_i^3 \cdot a_i \cdot \tilde{a}_i = \lambda_i \cdot (1 - \lambda_i a_i) \cdot (1 - \lambda_i \tilde{a}_i) .$$

Using independence once more we find

$$\mathbb{E}_{\mathbf{X},\tilde{\mathbf{X}}}[\mathbf{B}_{ii}] \geq \lambda_i \cdot \mathbb{E}_{\mathbf{X},\tilde{\mathbf{X}}}[(1 - \lambda_i a_i) \cdot (1 - \lambda_i \tilde{a}_i)] .$$

We proceed as in the proof of Lemma C.7 in Zou et al. (2021a). Recall that

$$(1 - \lambda_i a_i) = \frac{1}{1 + \lambda_i \langle z_i, \mathbf{A}_{-i}^{-1} z_i \rangle}$$

and for all $k$

$$\langle z_i, \mathbf{A}_{-i}^{-1} z_i \rangle \leq c \cdot \frac{n}{\lambda_{k+1} \rho_k} ,$$

for some $c > 0$, with high probability. Concluding as in Zou et al. (2021a) and using independence finishes the proof.

$\square$

*Proof of Proposition B.3.* Setting $w_m(\lambda) = \mathbf{H}^{1/2} \Pi_m(\lambda) w^*$ (see Definition B.1), we decompose the bias as

$$\mathrm{Bias}(\overline{w}_n^{\mathrm{RR}}(\lambda)) = \mathbb{E}\left[ \widehat{\mathrm{Bias}}(\overline{w}_n^{\mathrm{RR}}(\lambda)) \right]$$

$$= \mathbb{E}\left[ \left\| \frac{1}{M} \sum_{m=1}^{M} w_m(\lambda) \right\|^2 \right]$$

$$= \frac{1}{M^2} \mathbb{E}\left[ \mathrm{Tr}\left[ \left( \sum_{m=1}^{M} w_m(\lambda) \right) \otimes \left( \sum_{m'=1}^{M} w_{m'}(\lambda) \right) \right] \right]$$

$$= \frac{1}{M^2} \sum_{m=1}^{M} \mathbb{E}[\mathrm{Tr}[w_m(\lambda) \otimes w_m(\lambda)]] + \frac{1}{M^2} \sum_{m \neq m'} \mathbb{E}[\mathrm{Tr}[w_m(\lambda) \otimes w_{m'}(\lambda)]] . \tag{34}$$

We aim to find a lower for the above expression. Since

$$\sum_{m=1}^{M} \mathbb{E}[\text{Tr}[w_m(\lambda) \otimes w_m(\lambda)]] \succeq 0$$

we proceed to lower bound the second term in equation 34 . Setting

$$\mathbf{B}_{m,m'} := \Pi_m(\lambda) \circ \mathbf{H} \circ \Pi_{m'}(\lambda)$$

for $m, m' \in [M]$ we may write

$$
\begin{aligned}
\text{Bias}(\overline{w}_n^{\text{RR}}(\lambda)) &\geq \frac{1}{M^2} \sum_{m \neq m'} \mathbb{E}[\text{Tr}[w_m(\lambda) \otimes w_{m'}(\lambda)]] \\
&= \frac{1}{M^2} \sum_{m \neq m'} \mathbb{E}[\langle \mathbf{H} \circ \Pi_m(\lambda) w^*, \Pi_{m'}(\lambda) \rangle] \\
&= \frac{1}{M^2} \sum_{m \neq m'} \mathbb{E}[\langle \mathbf{B}_{m,m'} w^*, w^* \rangle] \\
&= \frac{1}{M^2} \sum_{m \neq m'} \left( \sum_i \mathbb{E}[(\mathbf{B}_{m,m'})_{ii}](w_i^*)^2 + 2 \sum_{i>j} \mathbb{E}[(\mathbf{B}_{m,m'})_{ij}] w_i^* \cdot w_j^* \right).
\end{aligned}
$$ (35)

We now apply Lemma B.4 and follow the lines of the proof of Theorem C.8 in Zou et al. (2021a) to obtain for every $k$

$$
\begin{aligned}
\text{Bias}(\overline{w}_n^{\text{RR}}(\lambda)) &\geq \frac{1}{M^2} \sum_{m \neq m'} \sum_i \mathbb{E}[(\mathbf{B}_{m,m'})_{ii}](w_i^*)^2 \\
&\geq \frac{1}{cM^2} \sum_{m \neq m'} \sum_i \frac{\lambda_i \cdot (w_i^*)^2}{\left(1 + \frac{\lambda_i}{\lambda_{k+1}} \cdot \frac{n}{M\rho_k}\right)^2} \\
&= \frac{M-1}{cM} \sum_i \frac{\lambda_i \cdot (w_i^*)^2}{\left(1 + \frac{\lambda_i}{\lambda_{k+1}} \cdot \frac{n}{M\rho_k}\right)^2} \\
&\geq \frac{M-1}{cM} \cdot \left( \frac{M^2 \left(\lambda + \sum_{j>k_{\text{RR}}^*} \lambda_j\right)^2}{n^2} \cdot \|w^*\|_{\mathbf{H}_{0:k_{\text{RR}}^*}^{-1}}^2 + \|w^*\|_{\mathbf{H}_{k_{\text{RR}}^*:\infty}}^2 \right),
\end{aligned}
$$ (36)

for some $c > 1$. $\qquad\qquad\square$

### B.1.3   Lower Bound of Variance for DRR

**Proposition B.5** (Lower Bound of Bias for local RR). *Suppose $k_{\text{RR}}^* < \frac{n}{c'M}$, for some universal constant $c' > 1$. There exist constants $b, c > 1$ such that*

$$\text{Var}(\overline{w}_n^{\text{RR}}(\lambda)) \geq \frac{\sigma^2}{c} \left( \frac{k_{\text{RR}}^*}{n} + \frac{n}{M^2 \cdot (\lambda + \sum_{j>k_{\text{RR}}^*} \lambda_j)} \cdot \sum_{j>k_{\text{RR}}^*} \lambda_j^2 \right),$$

*where $k_{\text{RR}}^*$ is defined in equation 33.*

*Proof of Proposition B.5.* By definition of the variance, we may write

$$
\begin{aligned}
\operatorname{Var}(\overline{w}_n^{\mathrm{RR}}(\lambda)) &= \mathbb{E}\Big[\widehat{\operatorname{Var}}(\overline{w}_n^{\mathrm{RR}}(\lambda))\Big] \\
&= \mathbb{E}\left[\left\|\mathbf{H}^{1/2}\left(\frac{1}{M}\sum_{m=1}^{M}(\mathbf{X}_m^T\mathbf{X}_m+\lambda)^{-1}\mathbf{X}_m^T\epsilon_m\right)\right\|^2\right] \\
&= \frac{1}{M^2}\sum_{m,m'=1}^{M}\mathbb{E}_{\mathbf{X}}\Big[\operatorname{Tr}\Big[\mathbf{H}^{1/2}(\mathbf{X}_m^T\mathbf{X}_m+\lambda)^{-1}\mathbf{X}_m^T\mathbb{E}_{\epsilon_m}[\epsilon_m\otimes\epsilon_{m'}]\mathbf{X}_m(\mathbf{X}_{m'}^T\mathbf{X}_{m'}+\lambda)^{-1}\mathbf{H}^{1/2}\Big]\Big] \\
&= \frac{\sigma^2}{M^2}\sum_{m=1}^{M}\mathbb{E}_{\mathbf{X}}\Big[\operatorname{Tr}\Big[\mathbf{H}^{1/2}(\mathbf{X}_m^T\mathbf{X}_m+\lambda)^{-1}\mathbf{X}_m^T\mathbf{X}_m(\mathbf{X}_m^T\mathbf{X}_m+\lambda)^{-1}\mathbf{H}^{1/2}\Big]\Big] \\
&= \frac{1}{M^2}\sum_{m=1}^{M}\operatorname{Var}(\hat{w}_m^{\mathrm{RR}}(\lambda))\,,
\end{aligned}
$$

where the local variance is given by

$$
\operatorname{Var}(\hat{w}_m^{\mathrm{RR}}(\lambda)) = \sigma^2\mathbb{E}_{\mathbf{X}}\Big[\operatorname{Tr}\Big[\mathbf{H}^{1/2}(\mathbf{X}_m^T\mathbf{X}_m+\lambda)^{-1}\mathbf{X}_m^T\mathbf{X}_m(\mathbf{X}_m^T\mathbf{X}_m+\lambda)^{-1}\mathbf{H}^{1/2}\Big]\Big]\,.
$$

To lower bound the variance we utilize Theorem C.5 from Zou et al. (2021a) (see also Bartlett et al. (2020)) and obtain

$$
\begin{aligned}
\operatorname{Var}(\overline{w}_n^{\mathrm{RR}}(\lambda)) &\geq \frac{\sigma^2}{cM^2}\sum_{m=1}^{M}\left(\frac{M\cdot k_{\mathrm{RR}}^*}{n} + \frac{n}{M\cdot(\lambda+\sum_{j>k_{\mathrm{RR}}^*}\lambda_j)}\cdot\sum_{j>k_{\mathrm{RR}}^*}\lambda_j^2\right) \\
&= \frac{\sigma^2}{c}\left(\frac{k_{\mathrm{RR}}^*}{n} + \frac{n}{M^2\cdot(\lambda+\sum_{j>k_{\mathrm{RR}}^*}\lambda_j)}\cdot\sum_{j>k_{\mathrm{RR}}^*}\lambda_j^2\right)\,,
\end{aligned}
$$

provided $k_{\mathrm{RR}}^* < \frac{n}{c'M}$, for some universal constants $c, c' > 1$. $\qquad\square$

### B.1.4 Proof of Theorem 4.2

The proof follows by combining Proposition B.3 and Proposition B.5 with Lemma B.2.

### B.2 Upper Bound Excess Risk Tail-Averaged DSGD

**Theorem B.6** (Upper Bound Tail-averaged DSGD)**.** *Suppose Assumption 3.7 is satisfied. Let $\overline{\overline{w}}_{M_n}$ denote the tail-averaged distributed estimator with $n$ training samples and assume $\gamma < 1/\operatorname{Tr}[H]$. For arbitrary $k_1, k_2 \in [d]$*

$$
\mathbb{E}\big[L(\overline{\overline{w}}_M)\big] - L(w^*) = \operatorname{Bias}(\overline{\overline{w}}_M) + \operatorname{Var}(\overline{\overline{w}}_M)
$$

*with*

$$
\operatorname{Bias}(\overline{\overline{w}}_M) \leq \frac{c_b M^2}{\gamma^2 n^2}\cdot\left\|\exp\left(-\frac{n}{M}\gamma\mathbf{H}\right)w^*\right\|_{\mathbf{H}_{0:k_1}^{-1}}^2 + \|w^*\|_{\mathbf{H}_{k_1:\infty}}^2\,,
$$

$$
\operatorname{Var}(\overline{\overline{w}}_M) \leq c_v(1+R^2)\cdot\sigma^2\left(\frac{k_2}{n} + \frac{n\gamma^2}{M^2}\cdot\sum_{j>k_2}\lambda_j^2\right)\,,
$$

*for some universal constants $c_b, c_v > 0$.*

*Proof of Theorem B.6.* Utilizing equation 13 and Lemma 6.1 in Zou et al. (2021a), we have

$$
\text{Bias}(\overline{\overline{w}}_M) = \frac{1}{M} \sum_{m=1}^{M} \text{Bias}\left(\bar{w}_{\frac{n}{M}}^{(m)}\right)
$$

$$
\leq \frac{1}{M} \sum_{m=1}^{M} \frac{c_b M^2}{\gamma^2 n^2} \cdot \left\|\exp\left(-\frac{n}{M}\gamma\mathbf{H}\right)\right\|_{\mathbf{H}_{0:k_1}^{-1}}^{2} + \|w^*\|_{\mathbf{H}_{k_1:\infty}}^{2}
$$

$$
= \frac{c_b M^2}{\gamma^2 n^2} \cdot \left\|\exp\left(-\frac{n}{M}\gamma\mathbf{H}\right)w^*\right\|_{\mathbf{H}_{0:k_1}^{-1}}^{2} + \|w^*\|_{\mathbf{H}_{k_1:\infty}}^{2} ,
$$

for some universal constant $c_b > 0$.

For the variance, we utilize equation 15 and Lemma 6.1 in Zou et al. (2021a) once more to obtain

$$
\text{Var}(\overline{\overline{w}}_M) \leq \frac{1}{M^2} \sum_{m=1}^{M} \text{Var}\left(\bar{w}_{\frac{n}{M}}^{(m)}\right)
$$

$$
\leq c_v \frac{(1+R^2)\cdot\sigma^2}{M^2} \sum_{m=1}^{M} \left(\frac{k_2 M}{n} + \frac{n\gamma^2}{M}\cdot\sum_{j>k_2}\lambda_j^2\right)
$$

$$
= c_v(1+R^2)\cdot\sigma^2\left(\frac{k_2}{n} + \frac{n\gamma^2}{M^2}\cdot\sum_{j>k_2}\lambda_j^2\right),
$$

for some universal constant $c_v > 0$. $\qquad\square$

## B.3 Comparing DSGD with DRR

### B.3.1 Proof of Theorem 4.4

To prove Theorem 4.4 we derive conditions on $n_{\text{RR}}$ and $n_{\text{SGD}}$ such that the upper bound for the excess risk of $\overline{\overline{w}}_M$ for DSGD from Theorem 4.3 can be upper bounded by the lower bound of $\overline{w}_n^{\text{RR}}(\lambda)$ for DRR from Theorem 4.2, i.e. such that

$$
\frac{c_b M^2}{\gamma^2 n_{\text{SGD}}^2} \cdot \left\|\exp\left(-\frac{n_{\text{SGD}}}{M}\gamma\mathbf{H}\right)w^*\right\|_{\mathbf{H}_{0:k_{\text{RR}}^*}^{-1}}^{2} + \|w^*\|_{\mathbf{H}_{k_{\text{RR}}^*:\infty}}^{2}
$$

$$
\leq \frac{M^2\left(\lambda + \sum_{j>k_{\text{RR}}^*}\lambda_j\right)^2}{c n_{\text{RR}}^2} \cdot \|w^*\|_{\mathbf{H}_{0:k_{\text{RR}}^*}^{-1}}^{2} + \|w^*\|_{\mathbf{H}_{k_{\text{RR}}^*:\infty}}^{2} \tag{37}
$$

and

$$
c_v\left(1 + \frac{\|w^*\|^2}{\sigma^2}\right)\cdot\sigma^2\left(\frac{k_{\text{RR}}^*}{n_{\text{SGD}}} + \frac{n_{\text{SGD}}\gamma^2}{M^2}\cdot\sum_{j>k_{\text{RR}}^*}\lambda_j^2\right) \leq \frac{\sigma^2}{c}\left(\frac{k_{\text{RR}}^*}{n_{\text{RR}}} + \frac{n_{\text{RR}}}{M^2}\cdot\frac{\sum_{j>k_{\text{RR}}^*}\lambda_j^2}{(\lambda + \sum_{j>k_{\text{RR}}^*}\lambda_j)^2}\right). \tag{38}
$$

We start with equation 38. For

$$
c_v\left(1 + \frac{\|w^*\|^2}{\sigma^2}\right)\cdot\sigma^2\frac{k_{\text{RR}}^*}{n_{\text{SGD}}} \leq \frac{\sigma^2}{c}\frac{k_{\text{RR}}^*}{n_{\text{RR}}}
$$

to hold we need that

$$
C^* n_{\text{RR}} \leq n_{\text{SGD}} , \quad C^* := c_v \cdot c \cdot \left(1 + \frac{\|w^*\|^2}{\sigma^2}\right). \tag{39}
$$

To
$$c_v\left(1+\frac{||w^*||^2}{\sigma^2}\right)\cdot\sigma^2\frac{n_{\mathrm{SGD}}\gamma^2}{M^2}\cdot\sum_{j>k^*_{\mathrm{RR}}}\lambda_j^2 \leq \frac{\sigma^2}{c}\frac{n_{\mathrm{RR}}}{M^2}\cdot\frac{\sum_{j>k^*_{\mathrm{RR}}}\lambda_j^2}{(\lambda+\sum_{j>k^*_{\mathrm{RR}}}\lambda_j)^2}$$

to hold we need
$$n_{\mathrm{SGD}} \leq \frac{n_{\mathrm{RR}}}{C^*\cdot(C^*_\lambda)^2\gamma^2}\;,\quad C^*_\lambda := \lambda + \sum_{j>k^*_{\mathrm{RR}}}\lambda_j\;. \tag{40}$$

Finally, from equation 37 we need
$$\frac{c_b M^2}{\gamma^2 n_{\mathrm{SGD}}^2}\cdot\left\|\exp\left(-\frac{n_{\mathrm{SGD}}}{M}\gamma\mathbf{H}\right)w^*\right\|^2_{\mathbf{H}^{-1}_{0:k^*_{\mathrm{RR}}}} \leq \frac{M^2(C^*_\lambda)^2}{c n_{\mathrm{RR}}^2}\cdot||w^*||^2_{\mathbf{H}^{-1}_{0:k^*_{\mathrm{RR}}}}\;. \tag{41}$$

To ensure this, note that
$$\left\|\exp\left(-\frac{n_{\mathrm{SGD}}}{M}\gamma\mathbf{H}\right)w^*\right\|^2_{\mathbf{H}^{-1}_{0:k^*_{\mathrm{RR}}}} \leq e^{-\frac{n_{\mathrm{SGD}}}{M}\gamma\lambda_{k^*_{\mathrm{RR}}}}\cdot||w^*||^2_{\mathbf{H}^{-1}_{0:k^*_{\mathrm{RR}}}} \leq (1-\gamma\lambda_{k^*_{\mathrm{RR}}})\cdot||w^*||^2_{\mathbf{H}^{-1}_{0:k^*_{\mathrm{RR}}}}\;.$$

Hence, equation 41 is implied if
$$\frac{c_b}{\gamma^2 n_{\mathrm{SGD}}^2}(1-\gamma\lambda_{k^*_{\mathrm{RR}}}) \leq \frac{(C^*_\lambda)^2}{c n_{\mathrm{RR}}^2}\;,$$

being equivalent to
$$\frac{\sqrt{cc_b(1-\gamma\lambda_{k^*_{\mathrm{RR}}})}}{\gamma C^*_\lambda}n_{\mathrm{RR}} \leq n_{\mathrm{SGD}}\;. \tag{42}$$

Combining conditions equation 39, equation 40 and equation 42 we need
$$\max\left\{C^*,\frac{\sqrt{cc_b(1-\gamma\lambda_{k^*_{\mathrm{RR}}})}}{\gamma C^*_\lambda}\right\}\cdot n_{\mathrm{RR}} \leq n_{\mathrm{SGD}} \leq \frac{1}{C^*\cdot(C^*_\lambda)^2\gamma^2}\cdot n_{\mathrm{RR}}\;.$$

A short calculation shows that the condition
$$\gamma < \min\left\{\frac{1}{\mathrm{Tr}[H]},\frac{1}{\sqrt{c}C^*C^*_\lambda}\right\}$$

implies that
$$\max\left\{C^*,\frac{\sqrt{cc_b(1-\gamma\lambda_{k^*_{\mathrm{RR}}})}}{\gamma C^*_\lambda}\right\} \leq \frac{1}{C^*\cdot(C^*_\lambda)^2\gamma^2}\;.$$

This finishes the proof.

### B.3.2 Discussion

We give a more detail discussion about the sample complexities (SCs) of DSGD and DRR. In particular, we derive conditions under which the SCs are of the same order to ensure that
$$\mathbb{E}\left[L(\overline{\overline{w}}_{M_{n_{\mathrm{SGD}}}})\right] - L(w^*) \leq \mathbb{E}\left[L(\overline{w}^{\mathrm{RR}}_{n_{\mathrm{RR}}}(\lambda_{n_{\mathrm{RR}}}))\right] - L(w^*)\;.$$

Recall that in order to achieve this bound we would need
$$L_{\lambda_{n_{\mathrm{RR}}},\gamma}\cdot n_{\mathrm{RR}} \leq n_{\mathrm{SGD}} \leq L'_{\lambda_{n_{\mathrm{RR}}},\gamma}\cdot n_{\mathrm{RR}}\;,$$

for a suitable choice of the regularization parameter $\lambda_{n_{\mathrm{RR}}}$ and number of machines $M_{n_{\mathrm{SGD}}}$.

To ensure that $n_{\mathrm{RR}} \lesssim n_{\mathrm{SGD}}$ we need to require that

$$1 \lesssim L_{\lambda_{n_{\mathrm{RR}}},\gamma} = \max\left\{ C^*, \frac{\sqrt{c(1-\gamma\lambda_{k_{\mathrm{RR}}^*})}}{\gamma C^*_{\lambda_{n_{\mathrm{RR}}}}} \right\},$$

with $C^*_\lambda := \lambda + \sum_{j>k_{\mathrm{RR}}^*} \lambda_j$. Recall that

$$\gamma < \min\left\{ \frac{1}{\mathrm{Tr}[H]}, \frac{1}{\sqrt{c}C^* C^*_{\lambda_{n_{\mathrm{RR}}}}} \right\},$$

and that $1 - \gamma\lambda_{k_{\mathrm{RR}}^*} < 1$. A short calculation shows that

$$1 \lesssim \frac{\sqrt{c(1-\gamma\lambda_{k_{\mathrm{RR}}^*})}}{\gamma C^*_{\lambda_{n_{\mathrm{RR}}}}}$$

if

$$\gamma\left( \lambda_{n_{\mathrm{RR}}} + \sum_{j>k_{\mathrm{RR}}^*} \lambda_j \right) \lesssim 1 .$$

Furthermore, to ensure that $n_{\mathrm{SGD}} \lesssim n_{\mathrm{RR}}$ we have to require that

$$L'_{\lambda_{n_{\mathrm{RR}}},\gamma} = \frac{1}{C^* \gamma^2 (C^*_{\lambda_{n_{\mathrm{RR}}}})^2} \lesssim 1 .$$

This is satisfied if

$$1 \lesssim \gamma \cdot C^*_{\lambda_{n_{\mathrm{RR}}}} = \gamma\left( \lambda_{n_{\mathrm{RR}}} + \sum_{j>k_{\mathrm{RR}}^*} \lambda_j \right) .$$

We summarize our finding the following

**Corollary B.7.** *Suppose all assumptions of Theorem 4.4 are satisfied. If*

$$\gamma\left( \lambda_{n_{\mathrm{RR}}} + \sum_{j>k_{\mathrm{RR}}^*} \lambda_j \right) \simeq 1 \tag{43}$$

*holds, then the sample complexities of DSGD and DRR are of the same order, i.e.*

$$n_{\mathrm{SGD}} \simeq n_{\mathrm{RR}}$$

*and*

$$\mathbb{E}\left[ L(\overline{\overline{w}}_{M_{n_{\mathrm{SGD}}}}) \right] - L(w^*) \ \le \ \mathbb{E}\left[ L(\overline{w}^{\mathrm{RR}}_{n_{\mathrm{RR}}}(\lambda_{n_{\mathrm{RR}}})) \right] - L(w^*) .$$

**Example B.8** (Spiked Covariance Model)**.** *We show that condition equation 43 is satisfied in the spiked covariance model from Corollary 3.9 under a suitable choice for $\lambda_{n_{\mathrm{RR}}}$ and $M_{n_{\mathrm{SGD}}}$. Here, we assume that with*

$$M_{n_{\mathrm{SGD}}} = M_{n_{\mathrm{RR}}} \simeq n_{\mathrm{RR}}^{\frac{3-2r}{5-2r}} ,$$

*for $1/2 \le r \le 1$, see our discussion in Section 3.4 (comparison with DOLS). A short calculation shows that*

$$k_{\mathrm{RR}}^* \simeq \tilde{d} \simeq \left( \frac{n_{\mathrm{RR}}}{M_{n_{\mathrm{RR}}}} \right)^r \simeq n_{\mathrm{RR}}^{\frac{2r}{5-2r}} .$$

*Moreover, for $\lambda_{n_{\mathrm{RR}}} \simeq n_{\mathrm{RR}}^{-\zeta}$, $\zeta \geq 0$ and $\gamma = const.$, we have*

$$\gamma\left(\lambda_{n_{\mathrm{RR}}} + \sum_{j > k_{\mathrm{RR}}^*} \lambda_j\right) \simeq \gamma\left(n_{\mathrm{RR}}^{-\zeta} + 1\right) \simeq 1 \,.$$

*Hence, for a wide range of regularization, the condition equation 43 is met and the SCs of DSGD and DRR in the spiked covariance model are of the same order.*

## C   FURTHER NUMERICAL EXPERIMENTS

In this Section we collect further experimental results conducted on simulated data from Section 5.

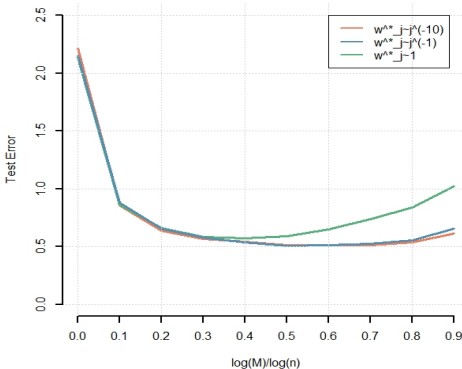

Figure 4: Test error for distributed ridgeless regression with $\lambda_j = j^{-2}$ for different sources $w^*$ as a function of $M = n^\alpha$, $\alpha \in \{0, 0.1, ..., 0.9\}$. The number of local nodes acts as a regularization parameter. We generate $n = 500$ i.i.d. training data with $x_j \sim \mathcal{N}(0, \mathbf{H})$ with mildly overparameterization $d = 700$.

We compare the sample complexity of optimally tuned full-averaged DSGD, tail-averaged DSGD and last-iterate DSGD with optimally tuned DRR for different sources $w^*$, see Figures 5, 6 and 6. Here, the data are generated as in Section 5 with $d = 200$, $\lambda_j = j^{-2}$ and $w_j^* = j^{-\alpha}$, $\alpha \in \{0, 1, 10\}$. The number of local nodes is fixed at $M_n = n^{1/3}$ for each $n \in \{100, ..., 6000\}$.

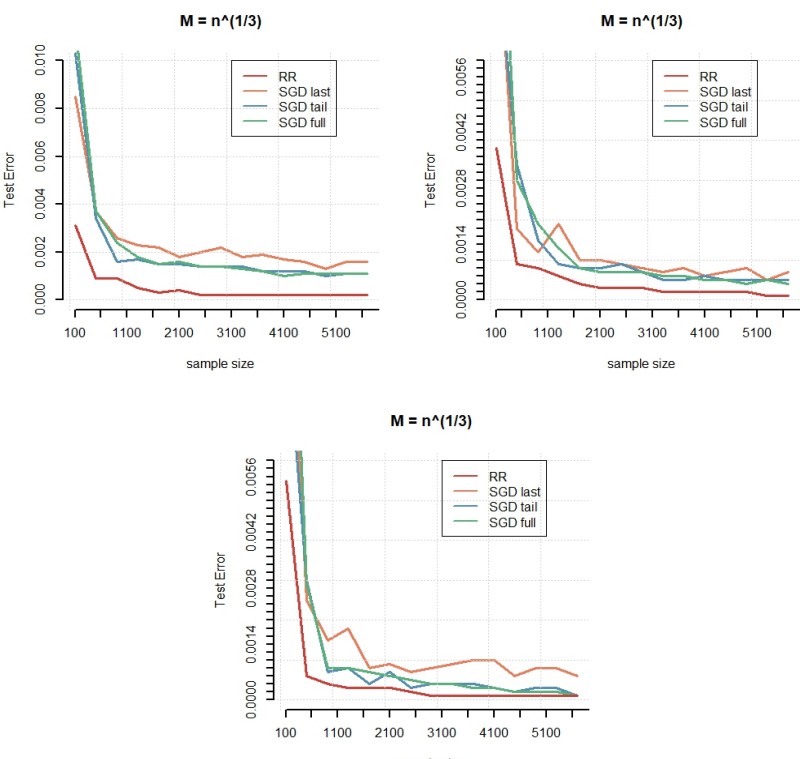

Figure 5: **Left:** $\lambda_j = j^{-10}$, $w_j^* = 1$ **Middle:** $\lambda_j = j^{-10}$, $w_j^* = j^{-1}$ **Right:** $\lambda_j = j^{-10}$, $w_j^* = j^{-10}$

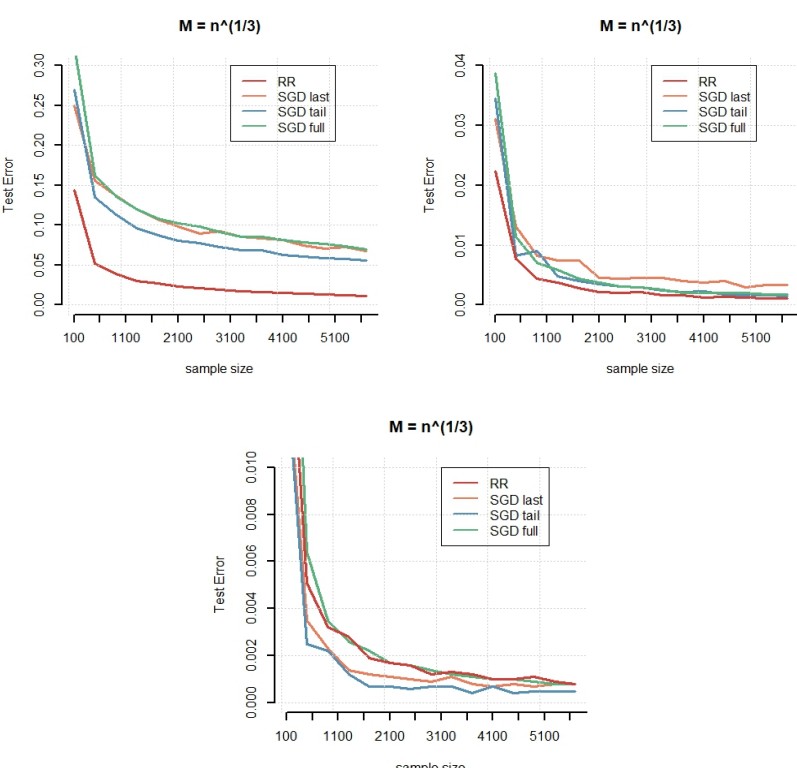

Figure 6: **Left:** $\lambda_j = j^{-2}$, $w_j^* = 1$ **Middle:** $\lambda_j = j^{-2}$, $w_j^* = j^{-1}$ **Right:** $\lambda_j = j^{-2}$, $w_j^* = j^{-10}$

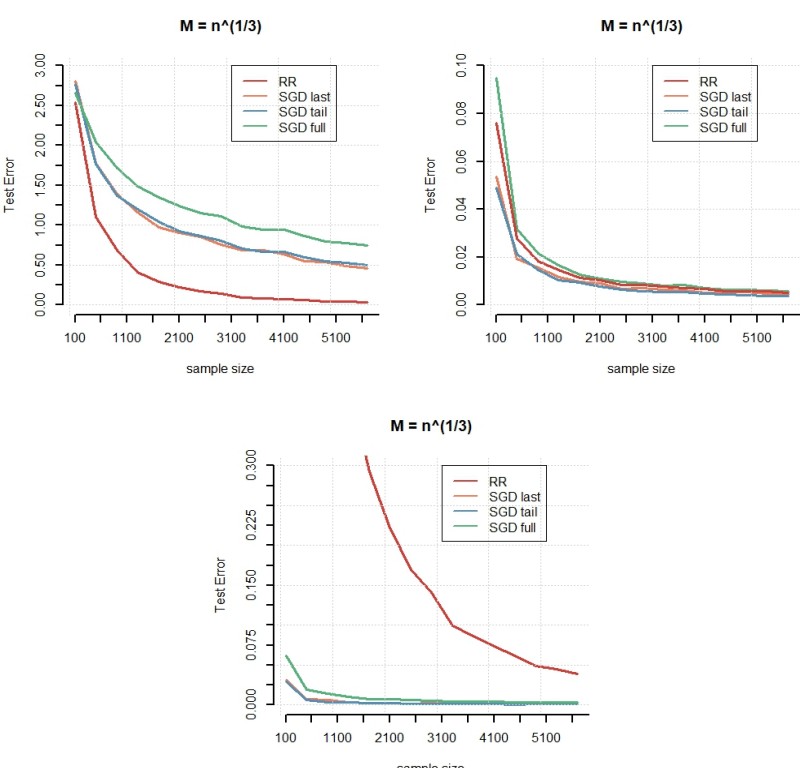

Figure 7: **Left:** $\lambda_j = j^{-1}$, $w_j^* = 1$ **Middle:** $\lambda_j = j^{-1}$, $w_j^* = j^{-1}$ **Right:** $\lambda_j = j^{-1}$, $w_j^* = j^{-10}$

