# OpenReview forum: "Distributed SGD in overparameterized Linear Regression"
_TMLR — Rejected by TMLR_

### Review · Reviewer_tXf8 · 2023-04-11

**Summary Of Contributions:**

In the overparametrised linear resgression setting, the authors analyse averaged constant stepsize local SGD (one iteration over the whole dataset) and provide an upperbound for the excess risk which depends on structural assumptions on the data. They then compute these upperbounds for two particular data distributions and compare these to those of distributed ridge regression. The sample complexity of distributed SGD is no worse than a constant factor of that of distributed ridge regression.

**Audience:**

Yes

**Claims And Evidence:**

No

**Requested Changes:**

Though I think the results can be interesting to some of TMLR's audience. I think they should be motivated in a totally different manner, the implicit regularisation / benign overfitting motivations are currently very misleading and wrong in my opinion. Thus I believe the introduction and motivations should be rewritten to be convincing.

**Strengths And Weaknesses:**

**Strengths:**
- from a point of view of the "form", the paper is well written: the setting, notations and results are clear and commented. The results seem sound and previously known results can be recovered.

**Weaknesses:**
I am unfortunately highly skeptical concerning many of the messages conveyed by this work and believe there is a huge gap between the original motivations and the considered setting. In my opinion, many of the statements in the paper are confusing, misleading or even wrong. Also, the major and relevant line of work concerning implicit regularisation is totally omitted: see *On the Implicit Bias in Deep-Learning Algorithms* from Gal Vardi and the references therein for a recent survey.

The authors start by introducing the concept of implicit regularisation. They then provide some references (Jain et al, Dieuleveut & Bach, Lin et al Varre et al etc) which have no link with implicit regularisation: one-pass SGD is known to directly optimise the true risk. The notion of implicit regularisation makes sense when reaching zero loss on the **empirical risk** (doing multi-pass SGD on it for example). In this case, since there are infinitely many global minimisers, it makes sense to look at which one the algorithm has converged to.

The authors then introduce distributed learning and local SGD, for which the motivation is clear. But the analysed algorithm in the paper is quite far from the actual local SGD algorithm since only one pass over the dataset is considered. This is never done in practice as the dataset is passed through many times. In fact, the actual local SGD algorithm would be much easier to analyse: if we initialise everything at $0$, all the updates remain in the span of the dataset and therefore we converge towards the OLS solution (which is the solution with the minimum $\ell_2$ norm), for which the benign overfitting phenomenon is well known from the Bartlett et al. paper (Benign overfitting in linear regression).

Therefore I don't understand the goal of the paper: if it is to understand the implicit regularisation of the actual multipass local SGD algorithm, then this is already provided by the results on the OLS estimator.  How does one-pass provide informations on the practical performances observed in practice?

Some sentences I disagree with:
- In the introduction: *In Zou et al. (2021b;a) it is shown that benign overfitting also occurs for SGD*, they consider one-pass SGD, for which we cannot overfit the dataset.
-  Page 3:  *It could be shown that benign overfitting occurs in this setting, i.e. the SGD estimator fits training data very well and still generalizes.*  Zou et al. (2021b), Wu et al. (2022) consider one-pass SGD, therefore very far from overfitting the train set. How do authors support the claim that *SGD estimator fits training data very well*? I have never seen practical settings where doing one-pass SGD leads to a solution close to zero-training error.
- Top of page 4: *Distributed learning in overparameterized linear regression is studied in Mücke et al. (2022) for the ordinary
least squares estimator (OLS), i.e. without any implicit or explicit regularization and with local interpolation.*  Choosing the OLS estimator is already an explicit choice of solution! there are many other interpolating solutions, by choosing the OLS solution the authors choose the minimum $\ell_2$ norm interpolating solution. Such sentences are very misleading.


Other comments:
- the *Introduction* section feels like a mix of an introduction and of related works. It would be much clearer to separate an actual introduction (with clear motivations), and a dedicated *related works* section.
- *It is generally believed that the optimization algorithm itself, e.g., stochastic gradient descent (SGD), implicitly regularizes such overparameterized models.*: I of course agree with this but it would be good to add at least a reference to support this claim
- speaking of linear regression: "*(to be considered as a reasonable approximation of neural network learning)*, how can we consider linear regression as a reasonable approximation of neural networks?!  The only way I can see this is through the NTK regime, which has been shown many times to be far from the actual performance regime which NNs work in.
- *Comparison to distributed ordinary least squares (DOLS)* paragraph:  the authors only compare the optimal number of nodes but not the optimal rates, how do they compare?
- Figures: the figures are quite disappointing and could highly be improved: the figures are squashed, the labels are hard to read: "w^*_j~j^-10"


Minor comments:
- First paragraph: *frameowrk*
- Contributions paragraph: the acronym DRR is not introduced before
- Notation paragraph: $\Vert w \Vert^2_A = \Vert A^{1/2} w \Vert$ should be $\Vert w \Vert^2_A = \Vert A^{1/2} w \Vert^2$
- equation (2): is this notation used anywhere in the paper?
- Theorem 3.5: parameter $\tau$ is not introduced
- Theorem 3.5: the definitions of  $\mathrm{Bias}(w)$ and  $\mathrm{Var}(w)$ are not given
- page 8: *Section5* -->  *Section 5*
- using the notation $\lambda$ for the ridge regression regularisation parameter and for the eigenvalues of $H$ can be confusing

---

### Review · Reviewer_nCK9 · 2023-05-24

**Summary Of Contributions:**

The paper considers the setting of distributed learning with SGD over multiple machines, where each machine runs one epoch on its local dataset and then sends the model to a central server, which then aggregates the models into the global model. For the setting of overparameterized linear regression, where each machine has i.i.d. data, they prove upper and lower bounds for the excess risk, under some assumptions that have been also used in prior work. They also show that under certain settings (if the number of machines
grows sufficiently slowly as compared to the total dataset size), the upper and the lower bounds match.

**Audience:**

Yes

**Claims And Evidence:**

Yes

**Requested Changes:**

I request the authors to address the concerns pointed out in the weakness section (in particular concerns 2, 3, and 4).

**Strengths And Weaknesses:**

Strengths:
1. Tightness of analysis: The upper bound and the lower bound provided in the paper match under certain conditions.
2. Insights into system design: The discussion in the paper on the upper and lower bounds provides some insight into how systems can be designed to minimize excess risk and achieve optimal sample complexity.
3. Comparison with distributed ridge regression: The paper shows that under some assumptions, the excess risk of distributed SGD is less than the excess risk of distributed ridge regression.

Weaknesses:
1. The setting of local SGD with single round of communication is motivated by use cases such as distributed training over user devices (like smartphones), because in that setup frequent communication could be expensive. In that case however, the dataset in different devices is not i.i.d. . On the other hand, the dataset is usually i.i.d. in use cases such as training in a data center, but then there the communication is cheap and we do not need a single round of communication.
2. The problem setup only considers a single epoch of local SGD (Section 2.2). This might be restrictive. Can the results be extended to the case where the local devices run multiple epochs, that is, the local SGD algorithm does multiple passes over the local datasets?
3. I am not able to grasp Assumption 3.6 (the fourth moment lower bound). In particular, for which kind of distributions would it hold?
4. Assumption 4.1. might be a strong assumption. If $z$ is some Gaussian random variable with identity covariance, and $x:=Az$, where $A$ is some matrix. Then $H = AA^T$, and $H^{-1}x = (AA^T)^{-1}Az$. In this case, I think $H^{-1}x$ does not have independent coordinates unless $(AA^T)^{-1}A$ is orthogonal.

---

### Review · Reviewer_FXvC · 2023-06-04

**Summary Of Contributions:**

The paper analyzes one-shot averaging in the overparameterized least squares setting. It derives a bias-variance decomposition and compares the algorithm against other distributed variants of ordinary least squares and ridge regression estimators. The paper also studies specific distributions and spectra to highlight the benefits of using one-shot averaging over these algorithms. Some experiments are provided to compare the baselines with each other.

**Audience:**

Yes

**Claims And Evidence:**

Yes

**Requested Changes:**

1) I encourage the authors to discuss in detail how the analysis differs from Zou et al. and if there are additional challenges in analyzing one-shot averaging (which have not been dealt with in the general convex literature.)

2) Discuss the relevance of the results in this paper when comparing them against known dimension-independent results. One can't appreciate this paper's results without highlighting which functions are missed by the dimension-independent analyses.

3) Experimentally, it would b good to add local SGD with multiple rounds of communication besides one-shot averaging.

**Strengths And Weaknesses:**

The math and the results are stated pretty clearly and rigorously. However, I have two major concerns about the paper:
1) It is unclear if the analysis here is a straightforward extension of Zou et al.'s analysis. Intuitively it is not hard to get the on-shot averaging analysis from the single machine analysis: one reduces the variance on the machines. This seems especially simple in the homogeneous setting. If there is not a significant technical challenge compared to Zou et al., then the paper doesn't have much of a contribution.

2) My biggest qualm with the paper is that it seems unfamiliar with very related results in the homogenous convex setting, which are dimension dependent (Woodworth et al. [a](https://arxiv.org/abs/2002.07839), [b](https://arxiv.org/abs/2102.01583)). In particular, the loss functions in this paper satisfy smoothness and bounded optimal property, which means we can apply  Theorem 1 from
[Woodowrth et al. a]((https://arxiv.org/abs/2002.07839)). This shows that one-shot averaging is optimal without looking at any specific spectral structure or distribution (up to acceleration). What is the setting where this result is insufficient and more fine-grained spectral analysis is needed? If there is no such setting, why would we po

---

### Decision · Action_Editors · 2023-07-28

**Recommendation:** Reject

**Comment:**

I can relate to the objections of the reviewers.
Although I think the paper has some merit and some results might be of interest to the community (note that the reviewers' comments focus mainly on the main result), it is not ideal if they are hidden and difficult to find. The paper should clearly state the differences from previous work and the new findings that have been brought to light.

It appears that some of the reviewer's concerns could have been addressed in a discussion. Unfortunately, the authors did not engage in a discussion and did not submit a revision. Therefore, I recommend rejection of the manuscript in its present form (as suggested by all three reviewers), with encouragement to submit a major revision at a later time.



**Audience:**

The reviewer's opinion is divided:
- on the one hand, the topic of the paper is clearly of interest to the TMLR community,
- on the other hand, however, the paper lacks a discussion of which aspects of the results might be of particular interest to the community (in light of previous work).
The reviewers asked for clarifications regarding the proof techniques (relation to (Zou et al.)), the general setting (one pass local SGD), and results in respect to prior works such as (Woodworth et al.).

**Claims And Evidence:**

The reviewers noted that all assumptions are clearly stated and that the mathematical claims and results proven in this paper appear correct.

However, reviewer tXf8 remarked that the discussion of prior work and related results should be strengthened (and sometimes carefully reworded) to avoid misinterpretations.

**Resubmission Of Major Revision:**

The authors may consider submitting a major revision at a later time.